# The myokine Fibcd1 is an endogenous determinant of myofiber size and mitigates cancer-induced myofiber atrophy

Flavia A. Graca[1,2], Mamta Rai [1,2,8], Liam C. Hunt[1,2,8], Anna Stephan[1,2], Yong-Dong Wang [3,4], Brittney Gordon[1,2,5], Ruishan Wang[1,2], Giovanni Quarato [6], Beisi Xu [3,7], Yiping Fan[3,7], Myriam Labelle [1,2] & Fabio Demontis [1,2✉]

Decline in skeletal muscle cell size (myofiber atrophy) is a key feature of cancer-induced wasting (cachexia). In particular, atrophy of the diaphragm, the major muscle responsible for breathing, is an important determinant of cancer-associated mortality. However, therapeutic options are limited. Here, we have used *Drosophila* transgenic screening to identify muscle-secreted factors (myokines) that act as paracrine regulators of myofiber growth. Subsequent testing in mouse myotubes revealed that mouse Fibcd1 is an evolutionary-conserved myokine that preserves myofiber size via ERK signaling. Local administration of recombinant Fibcd1 (rFibcd1) ameliorates cachexia-induced myofiber atrophy in the diaphragm of mice bearing patient-derived melanoma xenografts and LLC carcinomas. Moreover, rFibcd1 impedes cachexia-associated transcriptional changes in the diaphragm. Fibcd1-induced signaling appears to be muscle selective because rFibcd1 increases ERK activity in myotubes but not in several cancer cell lines tested. We propose that rFibcd1 may help reinstate myofiber size in the diaphragm of patients with cancer cachexia.

[1] Department of Developmental Neurobiology, St. Jude Children's Research Hospital, Memphis, TN, United States. [2] Solid Tumor Program, Comprehensive Cancer Center, St. Jude Children's Research Hospital, Memphis, TN, United States. [3] Department of Computational Biology, St. Jude Children's Research Hospital, Memphis, TN, United States. [4] Department of Cell and Molecular Biology, St. Jude Children's Research Hospital, Memphis, TN, United States. [5] Xenograft Core, St. Jude Children's Research Hospital, Memphis, TN, United States. [6] Department of Immunology, St. Jude Children's Research Hospital, Memphis, TN, United States. [7] Center for Applied Bioinformatics, St. Jude Children's Research Hospital, Memphis, TN, United States. [8] These authors contributed equally: Mamta Rai, Liam C. Hunt. ✉email: Fabio.Demontis@stjude.org

Skeletal muscle wasting, defined as the loss of muscle mass and strength, is a component of cachexia, a multifactorial syndrome characterized by systemic inflammation and weight loss, and it is a severe complication of many cancers[1–4]. It is estimated that cancer cachexia occurs in ~50% of tumors and that it is responsible for ~1/3 of cancer deaths[5,6]. Strikingly, even a modest cancer-induced decrease (~5–10%) in skeletal muscle mass reduces the functional capacity of the organism and increases the risk of mortality from cancer, whereas muscle hypertrophy is protective[7–12]. Specifically, preventing muscle mass loss in tumor-bearing mice improves prognosis and prolongs their survival even if cancer growth per se is not halted[8,9,13,14]. The mortality due to cachexia has been primarily attributed to wasting of the heart and of the diaphragm (the major muscle responsible for breathing)[9,11,15–18]. Despite the importance of diaphragm atrophy in determining cancer-associated mortality, only few therapeutic approaches are currently available. In particular, inspiratory muscle training has been found effective in improving diaphragm function in a number of disease conditions[19–23]. However, apart from this exercise regimen, no pharmaceutical therapies are available to impede diaphragm wasting.

At the cellular level, cancer-induced muscle wasting is primarily due to a decrease in the size of myofibers, the syncytial cells that compose the muscle[5,6,24]. Specifically, cancer cells and associated host stromal cells secrete a number of pro-inflammatory factors that promote myofiber atrophy by inducing catabolic signaling and muscle protein degradation[5,6,12,24–26]. Although an increase in tumor burden can lead to an increase in cachexia, there is a broad range of variability in cachexia progression and tumor size is only one of the factors (together with cancer type and host factors) that influences cachexia progression[27]. Importantly, the panel of cachectic cytokines produced by cancer cells and associated host stromal cells has been found to be a key determinant of cachexia. For example, cancer cachexia is abrogated in a lymphoma model when the translation of cachectic cytokines produced by the lymphoma is reduced, and this improves survival independently from neoplastic growth[28], as also found in other models[8,9,13,14]. Moreover, inflammatory cytokines can also be produced by immune cells, including those infiltrating the muscle, in response to tumor-derived signals[29,30], thus resulting in a systemic cytokine storm that is conducive to body wasting[31].

In addition to cachectic cytokines, the extent and outcome of cachexia is profoundly impacted by the responsiveness of target tissues to inflammatory/wasting signaling and/or the concurrent activation of compensatory pathways that mitigate cachexia-induced changes in target tissues[27]. In skeletal muscle, several signaling pathways have been found to regulate myofiber size in response to cachectic cytokines[5,6,12,24–26]. Interestingly, some of these pathways can also be modulated by muscle-secreted factors, i.e. myokines[32], and exerkines, i.e. signaling factors secreted by myofibers, muscle-infiltrating cells, as well as by other tissues in response to exercise[33]. Indeed, several myokines and exerkines regulate myofiber size in an autocrine/paracrine manner[34–36].

Initial evidence for the role of myokines in myofiber size regulation was found from the analysis of mice with postnatal knockout of the myokine myostatin, which resulted in myofiber hypertrophy[37–39]. Moreover, contraction-induced myokines (which are also referred to as exerkines[33]), such as irisin and decorin, were found to promote myofiber hypertrophy in response to exercise[40–43]. Other myokines are induced by muscle injury and promote muscle regeneration by acting on muscle satellite cells, differentiating myofibers, and/or muscle-infiltrating immune cells[44–49]. Importantly, a few myokines were also found to contrast disease-associated skeletal muscle wasting[50–54]. For example, the exerkine apelin reverses age-associated muscle

wasting (sarcopenia) and also promotes regeneration, mitochondrial biogenesis, and autophagy and impedes inflammatory pathways in myofibers[52]. However, apart from extensive efforts in developing myostatin inhibitors[55,56] and few other notable studies[52,57,58], the possibility of using recombinant growth-promoting myokines to treat disease-associated myofiber atrophy has not been explored exhaustively.

Here, we have used Drosophila developmental larval muscle growth[12,59,60] to screen >100 evolutionary-conserved myokines for their capacity to regulate myofiber size. Further testing of one of these myokines, secreted Fibcd1 (Fibrinogen C Domain-Containing Protein 1), determined its evolutionary-conserved role in myofiber size regulation in mice, and that recombinant Fibcd1 (rFibcd1) mitigates myofiber atrophy associated with cancer cachexia in the diaphragm muscle. Mechanistically, we find that Fibcd1 promotes MAPK/ERK signaling, which is necessary to preserve myofiber size and to respond to innervation in adult mice[58,61–64]. Interestingly, Fibcd1 increases ERK activity in myotubes but not in several cancer cells, although there are exceptions. This suggests that rFibcd1-based anti-wasting therapies may avoid the side effect of promoting ERK-driven cancer cell proliferation in the context of many cancers. We propose that interventions based on rFibcd1 may help reinstate myofiber size in the diaphragm of patients affected by cancer cachexia and by other disease states characterized by muscle wasting.

## Results

**RNAi and overexpression screening identifies evolutionary-conserved myokines that determine myofiber size in Drosophila.** Several myokines have been identified by transcriptomic and mass-spectrometry studies[32,36,65] but their role in muscle wasting remains largely unexplored. Signaling pathways that regulate muscle mass in mammals play similar roles also in Drosophila, as found for insulin receptor/FoxO signaling[12,59]. On this basis, we examined whether evolutionary-conserved myokines regulate the size of Drosophila larval body wall skeletal muscles, which each consist of a single myofiber. As previously shown[12,59,60], muscle-specific interventions that change the size of the body wall musculature correspondingly change the larval body size. On this basis, the larval size was used as primary screen readout to estimate the capacity of each myokine to regulate developmental muscle growth, which was followed by quantification of myofiber size. Overall, we followed an experimental pipeline (Fig. 1a, b) similar to the one we recently employed to screen for ubiquitin ligases[60] and transcription factors[66]. Specifically, the UAS/Gal4 system[67], the skeletal muscle-specific Mef2-Gal4 driver[59,68], and 508 UAS-RNAi transgenic fly stocks were used to knock down 111 evolutionary-conserved myokines that have strong muscle expression (RNA-seq FPKM values ≥4), and assess their role in myofiber growth (Fig. 1a–c). Whenever available, overexpression lines for the corresponding myokines were also tested.

In comparison to control RNAi lines for GFP and white, this muscle-specific RNAi screen and follow-up validation indicated that 31/508 (6.1%) RNAi lines induce myofiber atrophy (Fig. 1c, Supplementary Fig. 1, and Supplementary Data 1) similar to what is observed with overexpression of the transcription factor FoxO, indicating that these myokines are necessary to sustain the ~40-fold myofiber growth that occurs during larval development[12,59]. Conversely, similar to insulin receptor (InR) overexpression, 12/508 (2.3%) RNAi lines induced myofiber hypertrophy (Fig. 1c and Supplementary Data 1), indicating that these myokines normally limit developmental muscle growth, similar to mammalian myostatin[38,69]. Further testing via immunostaining confirmed that the size of representative skeletal muscles (VL3-

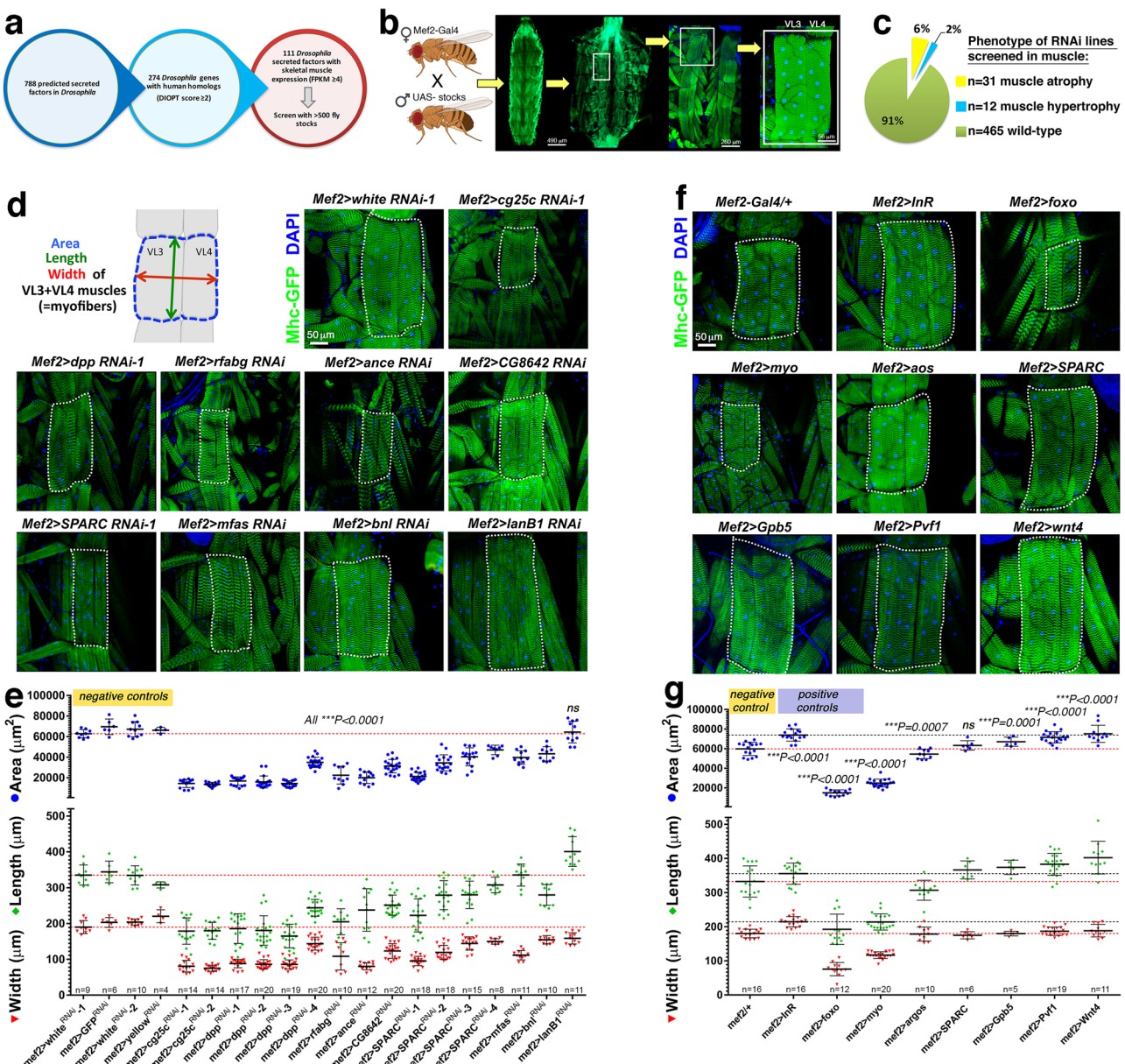

**Fig. 1 RNAi and overexpression screening identifies myokines that regulate myofiber size in *Drosophila* body wall skeletal muscles. a** Starting from 788 predicted secreted factors encoded by the *Drosophila* genome, 274 with human homology were selected based on a DIOPT homology score of ≥2. Of these, 111 secreted factors with substantial skeletal muscle expression (FPKM ≥ 4) were further chosen for screening with the UAS/Gal4 system and 508 transgenic stocks. **b** Transgenic RNAi or overexpression was driven in body wall skeletal muscle with *Mef2-Gal4* and muscle phenotypes were identified first based on overall larval body size and secondarily with dissections and analysis of the size of a stereotypical set of skeletal muscles, ventral longitudinal VL3 and VL4 muscles, each consisting of a single myofiber. **c** Screen results (see Supplementary Data 1 for a full report): 31 interventions induced myofiber atrophy (6.1%), 12 led to myofiber hypertrophy (2.4%), whereas 465 led to no phenotype (91.5%). **d** Representative images of RNAi interventions that induce myofiber atrophy, compared to negative control RNAi. **e** Quantitation of the myofiber area, width, and length indicates a significant decrease; mean values ± SD and the precise n(VL3 + VL4 muscles from independent larvae) are reported in the figure; *P* < 0.001. Statistical analysis was done by using two-way ANOVA with Dunnett's multiple comparison test. **f** Representative images of muscle size phenotypes induced by myokine overexpression, compared to a negative control (+; no transgene) and to positive controls (*foxo* and *Insulin Receptor* overexpression). **g** Quantitation of the myofiber area, width, and length indicates the significant induction of myofiber atrophy and hypertrophy by overexpression of the myokines indicated; mean values ± SD; the n(VL3 + VL4 muscles from independent larvae) is reported in the figure; *P* < 0.001. Statistical analysis was done by using two-way ANOVA with Dunnett's multiple comparison test. Source data and complete statistical analyses are provided in the Source Data file.

VL4) is indeed modulated by RNAi interventions that induce atrophy and hypertrophy (Fig. 1d). Importantly, transgene expression with this *Mef2-Gal4* does not affect the number of syncytial nuclei[59], indicating that the changes in myofiber size here observed do not arise from altered myoblast fusion.

Among the myokines that scored, a first class of atrophic phenotypes was induced by RNAi lines for growth factors, which indicates that they are necessary for myofiber growth in an autocrine/paracrine manner. For example, 4 different RNAi lines for the myostatin-binding protein SPARC reduced myofiber size

(Fig. 1d, e), suggesting that SPARC normally prevents atrophy by binding to and inhibiting TGF-β ligands such as myostatin, as found in mice[70]. Moreover, muscle-specific RNAi for the BMP2/4 homolog *dpp* also induced myofiber atrophy (Fig. 1d, e), presumably via the capacity of BMP ligands to antagonize TGF-β signaling and promote myofiber hypertrophy[71].

Lastly, RNAi for the FGF homolog *bnl* decreased myofiber size (Fig. 1d, e), possibly by limiting the development of the muscle-associated trachea, which delivers oxygen to sustain muscle growth, similar to the mammalian vasculature[72], whereas over-expression of the PDGF/VEGF homolog *Pvf1* induced myofiber hypertrophy (Fig. 1f, g). Other growth factors included *Wnt4* and *Gpb5* (homologous to WNT9 and GPHB5, respectively), which induced hypertrophy when overexpressed. Conversely, over-expression of *aos* (an inhibitor of EGF receptor signaling) and *myoglianin* (homologous to myostatin/GDF11[73]) induced atrophy (Fig. 1f, g).

A second category of atrophy inducers consisted of RNAi for the extracellular matrix proteins *cg25c* and *mfas* (Fig. 1d, e), respectively homologous to collagen COL4A1 and to the collagen-binding protein TGFBI. Consistently, COL4A1 mutations cause myopathy in humans[74] whereas TGFBI loss reduces developmental myofiber growth in zebrafish[75]. Interestingly, RNAi for *lanB1*, homologous to laminin LAMB2, did not significantly affect myofiber size but rather conferred an elongated shape to myofibers (Fig. 1d, e). Other inducers of myofiber atrophy included RNAi for the angiotensin-converting enzyme 1/2 *ance*, for the apolipoprotein *rfabg*, and for *CG8642*, a myokine homologous to mouse Fibrinogen C Domain-Containing Protein 1 (Fibcd1). Altogether, this screen has identified evolutionary-conserved myokines that regulate myofiber size.

**siRNAs for mouse Fibcd1 induce atrophy in mouse C2C12 myotubes**. The growth-promoting myokines identified in the *Drosophila* screen are evolutionary conserved. On this basis, we next selected few mouse orthologs for these *Drosophila* myokines to test whether they regulate the size of cultured mouse C2C12 myotubes. For these studies, we first assessed the capacity of siRNAs targeting these myokines to impact myoblast fusion, and subsequently whether siRNAs transfected post-fusion induce atrophy in myotubes. Analysis at day 4 of myogenic differentiation revealed that siRNAs for Bmp1 reduced myoblast fusion whereas siRNAs targeting Wnt9a, Tgfbi, Sparc, and Fibcd1 did not (Supplementary Fig. 2a, b).

When transfected post-fusion into C2C12 myotubes, siRNAs for Tgfbi, Sparc, and Fibcd1 induced atrophy, compared to NT control siRNAs (Supplementary Fig. 2a, c). Importantly, via a decline in *Fibcd1* mRNA levels, Fibcd1 siRNAs induced the strongest atrophy and also worsened starvation-induced atrophy (Fig. 2 and Supplementary Fig. 2c, d). On this basis, we focused further studies on Fibcd1.

**Secreted Fibcd1 rescues myotube atrophy induced by siRNAs for Fibcd1**. In *Drosophila*, the Fibcd1 ortholog *CG8642* encodes for a secreted protein that contains a fibrinogen-like domain. In mice, several splice variants arise from the *Fibcd1* gene. Specifically, the full-length mRNA encodes for a transmembrane version of Fibcd1 (459 amino acids, FL, consisting of exons 1–7), which contains an extracellular fibrinogen domain and has been reported to work as a chitin receptor in epithelia[76,77]. In addition, a short Fibcd1 splice variant lacks the transmembrane region but carries the fibrinogen domain (202 amino acids, SH, consisting of exons1,5-7) and may therefore be secreted (Fig. 2a). To test this hypothesis, C-terminally flag-tagged short and long versions of mouse Fibcd1 were transfected into HEK293 cells and the

amount of Fibcd1 recovered from the supernatant was probed with anti-flag antibodies. Both short (SH) and long (FL) versions of Fibcd1 were detected in similar amounts in transfected cells, compared to empty-vector (EV) controls. However, there was a ~38 kDa Fibcd1 fragment that was recovered from the culture medium of HEK293 cells transfected with full-length Fibcd1 (FL) but not from that of cells transfected with the short version of Fibcd1 (SH). These findings indicate that transmembrane Fibcd1 is cleaved C-terminally to generate a secreted Fibcd1 that includes the fibrinogen domain (Fig. 2b).

On this basis, we next tested whether secreted Fibcd1 rescues the myotube atrophy induced by Fibcd1 siRNAs. For these studies, we produced a ~38 kDa recombinant mouse Fibcd1 (rFibcd1; 282 C-terminal amino acids of FL Fibcd1) similar to the extracellular fragment released by proteolytic processing of Fibcd1. Treatment with rFibcd1 rescued the myotube atrophy induced by Fibcd1 siRNAs, indicating that the myotube atrophy induced by Fibcd1 siRNAs is indeed an RNAi on-target effect, and that rFibcd1 is sufficient to reinstate myofiber size (Fig. 2c).

**rFibcd1 rescues myotube atrophy induced by cachectic cytokines**. Cancer cells secrete inflammatory signaling factors that induce myofiber atrophy[5,6,12,24,25]. Because we have found that Fibcd1 is a myokine that preserves myofiber size, we next tested whether rFibcd1 can resolve myofiber atrophy induced by cachectic cytokines. To test this hypothesis, we treated C2C12 myotubes with IL-6 (20 ng/mL), LIF (20 ng/mL), and TNF-α (100 ng/mL), which are known to be upregulated in cancerous states and lead to muscle mass loss[25]. As expected based on previous reports[78–80], in vitro treatment with IL-6, LIF, and TNF-α induced significant atrophy of C2C12 myotubes. However, treatment with rFibcd1 significantly reduced the atrophy induced by IL-6, LIF, and TNF-α (Fig. 2d). Altogether, these findings suggest that Fibcd1 mitigates myotube atrophy induced by cachectic cytokines.

**rFibcd1 rescues myotube atrophy by promoting ERK signaling in muscle cells**. To examine what signaling pathways are modulated by Fibcd1 and are responsible for its effects on myotube size determination, we examined the effect of Fibcd1 siRNAs and rFibcd1 treatments on the activity of several signaling pathways with a panel of phospho antibodies. The addition of rFibcd1 to the cell culture medium of mouse C2C12 myotubes increased P-p42 and P-p44 (ERK1/2) MAPK levels after 30 min from stimulation (Fig. 3a), whereas no substantial changes were found in the activity of other pathways including JNK and p38 MAPK (Supplementary Fig. 3a).

ERK signaling is necessary for the maintenance of skeletal muscle mass[63] and its inhibition prevents IGF-I-induced myofiber hypertrophy[64]. Moreover, ERK has been shown to be necessary to maintain adult myofiber size and their neuromuscular junctions[58,61,62]. Mechanistically, ERK can preserve myofiber size via phosphorylation and inactivation of GSK3β[81,82] and by promoting protein synthesis via activation of the MAPK-interacting kinase MNK1/2, which in turn phosphorylates the translation initiation factor eIF4E[81,83,84].

On the basis of these studies, Fibcd1 may modulate myofiber size at least in part via ERK signaling.

To test this hypothesis, C2C12 myotubes were treated with Fibcd1 siRNAs, rFibcd1, and/or Pyrazolylpyrrole, a pharmacological inhibitor of ERK[35,85,86]. By using low doses of Pyrazolyl-pyrrole (2.5 ng/mL), we found that Pyr has no major impact on myotube size in normal conditions (Fig. 3b). However, Pyrazolylpyrrole prevents rFibcd1 from rescuing myotube atrophy induced by Fibcd1 siRNAs, indicating that rFibcd1 reinstates myotube size via its capacity to increase ERK activity (Fig. 3b).

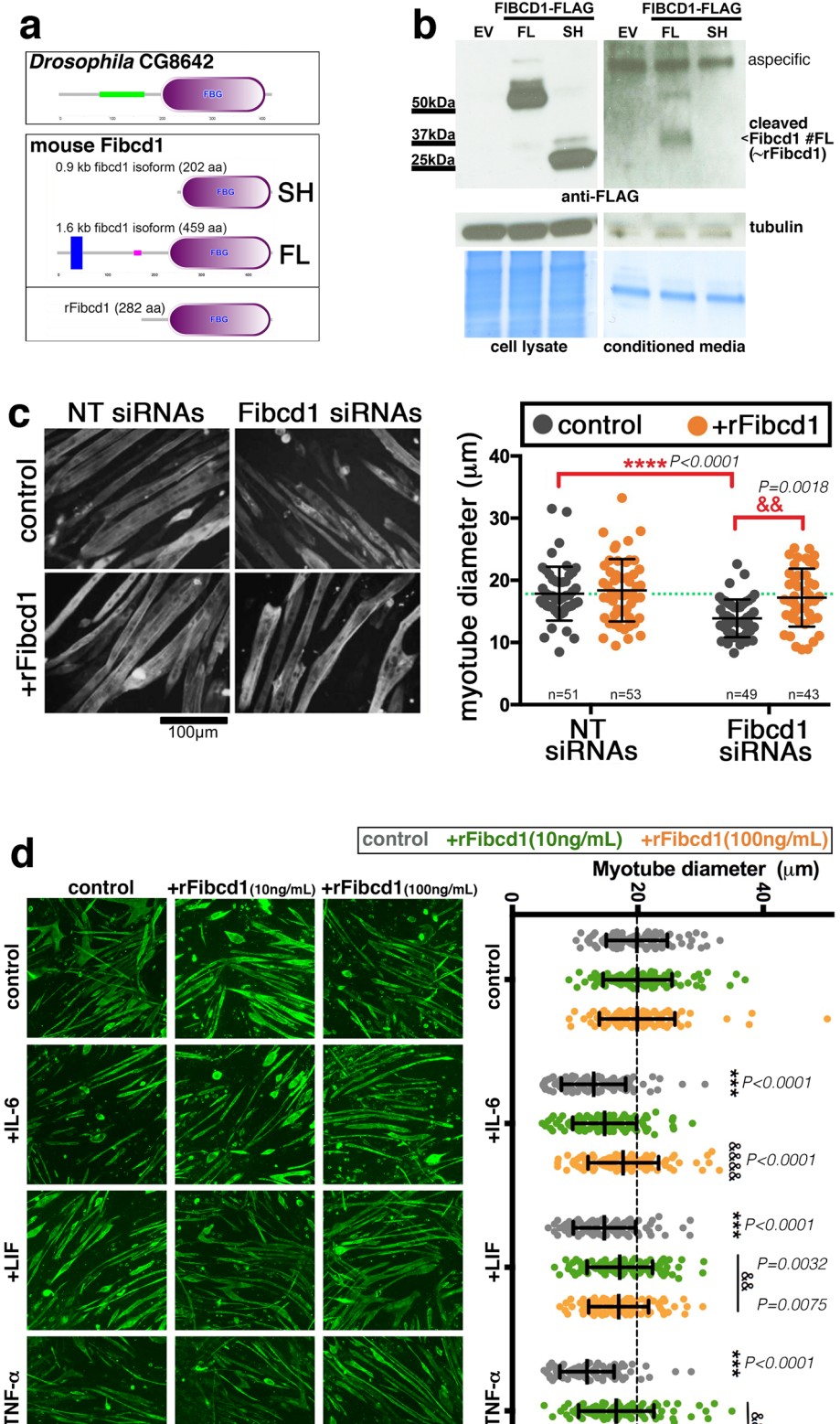

**rFibcd1 does not promote ERK signaling in a set of cancer cells**. ERK is a major driver of tumorigenesis[87] and therefore interventions that combat muscle wasting via ERK signaling may be hindered by the side effect of promoting cancer cell proliferation[88]. On this basis, we next examined whether rFibcd1 increases P-ERK levels also in cancer cells. To this purpose, we examined P-ERK levels in Lewis lung carcinoma (LLC)[89], 4T1 breast cancer[90,91], and Saos-2 osteosarcoma cells[92]. Different from C2C12 myotubes, western blot analyses indicated that rFibcd1 does not promote ERK signaling in these cancer cells, with the exception of Saos-2 cells treated with high levels of rFibcd1 (Fig. 3c). To further test whether rFibcd1 induces ERK

**Fig. 2 siRNAs for mouse Fibcd1 induce mouse C2C12 myotube atrophy. a** Transmembrane (FL) and short (SH) versions of the mouse Fibcd1 protein (orthologous to *Drosophila* Fibcd1/*CG8642*) are encoded by the mouse genome. **b** Western blot of HEK293 cell lysates and cell culture supernatants 2 days after transfection with either an empty vector (EV) or with a vector encoding C-terminal Flag-tagged full length (FL) or short (SH) mouse Fibcd1. Expression of FL and SH is detected at similar levels in cell lysates. However, although neither FL nor SH are detected in the culture medium, a C-terminal ~38 kDa fragment of Fibcd1 derived from the proteolytic processing of FL Fibcd1 is detected in the cell culture supernatant. Coomassie blue staining is shown as loading control. A recombinant Fibcd1 protein (rFibcd1) that resembles the cleaved Fibcd1 FL fragment has been generated **a**. **c** NT or Fibcd1 siRNAs transfection into mouse C2C12 myotube-enriched cultures for 48 h, followed by treatment for further 24 h with rFibcd1 or a vehicle control. Representative images of myotubes stained for myosin heavy chain are shown. Scale bar, 100 μm. Measurement of myotube width indicates that Fibcd1 siRNAs induce atrophy and that myotube size is rescues by rFibcd1. Data are mean ± SD with the precise n indicated in the figure; ****$P < 0.0001$ and &&$P < 0.01$, compared to the indicated control; $P$ values were determined by two-way ANOVA with Sidak's multiple comparisons test. See also Supplementary Fig. 2. **d** Myotube-enriched cultures treated for 48 h with cachectic cytokines (IL-6 at 20 ng/mL, LIF at 20 ng/mL, and TNF-α at 100 ng/mL) and either rFibcd1 at 10 and 100 ng/mL or vehicle-alone control at the same time of the cytokine treatment. Representative images of myotubes stained for myosin heavy chain (green) are shown. Scale bar, 200 μm. Measurement of myotube width indicates that rFibcd1 rescues atrophy induced by cachectic cytokines. Data are mean ± SD; $n = 101$ myotubes/group. ***$P < 0.001$ compared to control; &$P < 0.05$, &&$P < 0.01$, and &&&&$P < 0.0001$ compared to control within each cytokine group. Statistical analysis was done by using two-way ANOVA with Sidak's multiple comparisons test. Source data are provided in the Source Data file.

signaling in cancer cells, we probed a panel of osteosarcoma, colorectal adenocarcinoma, melanoma, and breast cancer cell lines. In most cases, rFibcd1 induced only marginal changes in P-ERK levels. However, there were also some notable exceptions: rFibcd1 increased P-p44/42 levels in 143B-Luc osteosarcoma and 67NR breast cancer cell lines (Fig. 3d and Supplementary Fig. 3b). Altogether these findings suggest that rFibcd1 may combat myofiber atrophy by inducing ERK signaling in muscle but not in cancer cells. However, this is not universally true because Saos-2, 143B-Luc, and 67NR breast cancer cell lines responded to rFibcd1. Therefore, while rFibcd1 may provide an effective therapy for treating cachexia-induced myofiber atrophy without fueling cancer growth in the context of many cancers, this may not be generally applicable to all cancer types.

Recombinant Fibcd1 primarily consists of a structural domain related to fibrinogen, which is known to bind to and signal via specific combinations of α and β integrins[93–95]. On this basis, rFibcd1 may bind to integrin receptors similar to fibrinogen (Fig. 3e) and this may represent a mechanism by which Fibcd1 promotes MAPK phosphorylation, a known readout of integrin signaling[95].

To examine whether α/β integrins known to bind to fibrinogen, including Itgα2b[93–95], are necessary for rFibcd1-induced ERK phosphorylation, we first examined their differential expression in C2C12 muscle cells versus rFibcd1-non-responsive cancer cells. RNA-seq indicates that whereas integrin β2 and β3 expression occurs across cell lines, *Itgα2b* is expressed in muscle but not in cancer cells (Fig. 3f). Analysis of *Itgα2b* expression from larger datasets further confirmed that Itgα2b is poorly or not expressed in many pediatric and adult cancers (Supplementary Fig. 4). Subsequent qRT-PCR analysis indicates that *Itgα2b* is not expressed in rFibcd1-non-responsive cancer cells (4T1, E0771, Ep5). Conversely, the rFibcd1-responsive cancer cell line 67NR (Fig. 3d and Supplementary Fig. 3b) displays *Itgα2b* expression levels similar to those of C2C12 myotubes (Fig. 3f), suggesting that Itgα2b might be a receptor for rFibcd1.

To test whether Itgα2b is indeed required for Fibcd1-mediated P-ERK induction, C2C12 myotubes were treated with Itgα2b siRNAs (Supplementary Fig. 3c). Compared to control NT siRNAs, Itgα2b siRNAs reduced P-ERK levels both at steady state and upon rFibcd1 administration (Fig. 3g), indicating that *Itgα2b* expression is needed for the optimal response of muscle cells to rFibcd1.

**rFibcd1 rescues LLC cancer-induced myofiber atrophy in the diaphragm muscle of mice**. To test the therapeutic potential of rFibcd1 in vivo, we employed an established model of cancer cachexia based on the subcutaneous injection of Lewis Lung Carcinoma (LLC) cells[60,96,97].

For these experiments, ~$10^6$ LLC cells were injected subcutaneously in each flank and allowed to grow for ~3 weeks. One week before reaching the endpoint, tumor-bearing mice were randomly allocated to receive 3 intraperitoneal injections of rFibcd1 (every other day; 1 mg/kg of rFibcd1 in PBS with 1%BSA) or mock (PBS with 1%BSA). Cohorts of isogenic C57BL/6 J control mice devoid of tumors were similarly injected with either rFibcd1 or PBS.

A key determinant of cachexia-associated mortality is wasting of the diaphragm, the muscle necessary for respiration[11,98]. Therefore, interventions that preserve the size of diaphragm myofibers are crucial for reducing mortality associated with cachexia and other critical illnesses[99]. On this basis, we have tested whether rFibcd1 rescues cancer-induced atrophy of diaphragm myofibers and found that this is the case (Fig. 4). Specifically, LLC cancer cells induce atrophy of all myofiber types (1, 2a, and 2x/2b) but this is rescued by intraperitoneal rFibcd1 injection (Fig. 4a, b), as indicated by the Feret's minimal diameter, a geometrical parameter indicative of myofiber size[100]. However, rFibcd1 injection does not affect myofiber size in wild-type (non-cachectic) conditions (Fig. 4a, b), indicating that Fibcd1 maintains steady-state myofiber size but does not induce hypertrophy.

Because myofiber types have different defining sizes, the reduction in myofiber atrophy due to rFibcd1 may be explained via its capacity to promote a shift from type 1 to type 2a or 2x/2b myofibers, which are bigger[101]. However, immunostaining of diaphragm muscles with antibodies for distinct myosin heavy chain isoforms revealed that there are no changes in the relative abundance of different myofiber types present in the diaphragm of mice treated with rFibcd1 versus controls (Fig. 4c).

We next analyzed the distribution of myofiber sizes present in the diaphragm. For each myofiber type, LLC cancer cells induced a shift towards smaller sizes, which was rescued by rFibcd1 (Fig. 4d–g).

The capacity of rFibcd1 to limit myofiber atrophy in the diaphragm of cancer-bearing mice arises via its local action in attenuating cachectic changes in the diaphragm muscle. Systemic effects of rFibcd1 on body wasting and/or on cancer cells are unlikely given that rFibcd1 is injected locally in the peritoneal cavity. Indeed, body weight and tumor-free body mass declined in mice injected with LLC cancer cells but did not differ in response to intraperitoneal rFibcd1 injection (Fig. 4h–i), indicating that intraperitoneal rFibcd1 is active on the diaphragm, which is located in the peritoneal cavity, but not systemically. Consistently,

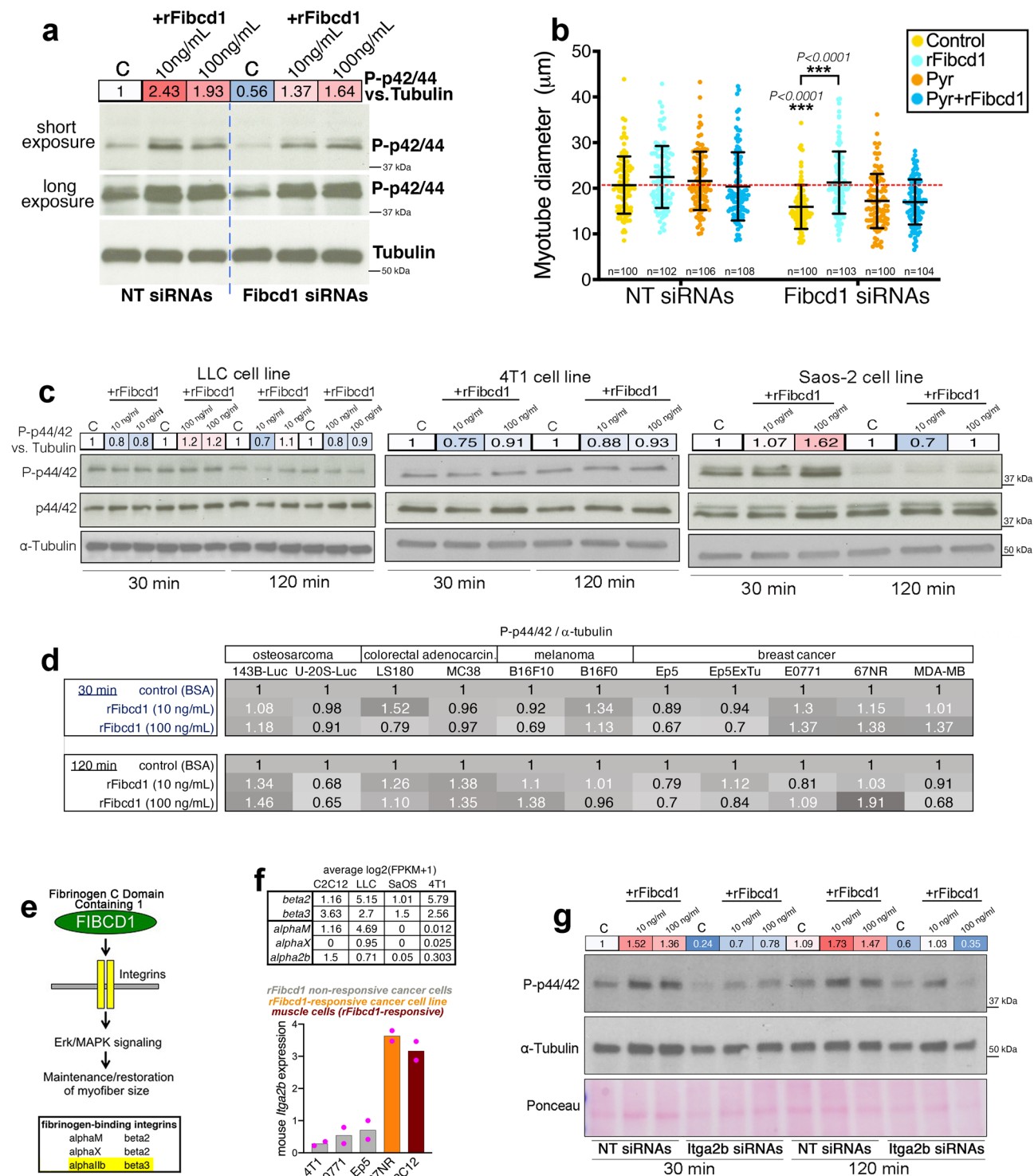

LLC tumor mass was similar in mice treated with rFibcd1 versus controls (Fig. 4j). Altogether, these findings indicate that local injection of rFibcd1 resolves atrophy of diaphragm myofibers induced by cancer cachexia and that this is not due to an effect of rFibcd1 on tumor burden.

**rFibcd1 rescues muscle transcriptional changes induced by LLC cancers.** We next examined whether rFibcd1 rescues the molecular changes associated with cachexia-induced myofiber atrophy in the diaphragm. In parallel with the preservation of

myofiber size (Fig. 4), RNA-sequencing revealed that LLC-induced cachexia is characterized by an increase in the expression of many genes whereas relatively few have reduced expression (Fig. 4k). However, rFibcd1 significantly mitigates LLC-induced transcriptional changes in the diaphragm muscle, compared to mock injections (Fig. 4k).

Further analysis of 247 significantly-regulated genes in LLC versus LLC + rFibcd1 (p < 0.05 and −0.5 > log2R > 0.5) revealed that several categories of genes involved in immunity and inflammation are overrepresented among LLC-upregulated genes (Fig. 4l), including the chemokine *Cxcl13* (Fig. 4m) and receptors

**Fig. 3 rFibcd1 preserves myotube size via inducing ERK signaling but does not activate this pathway in some cancer cells. a** Transfection of C2C12 myotubes for 48 h with Fibcd1 siRNAs reduces ERK signaling, as indicated by lower P-p44/42 MAPK compared to NT siRNAs. A 30-min treatment with rFibcd1 rescues P-p44/42 MAPK. See also Supplementary Fig. 3a. **b** NT or Fibcd1 siRNAs transfection into mouse C2C12 myotube-enriched cultures for 48 h, followed by treatment for further 24 h with Pyrazolylpyrrole (ERK inhibitor) at 2.5 ng/mL, indicates that ERK is needed for the preservation of myotube size by rFibcd1. Data are mean ± SD, with n indicated in the figure, and ***$P < 0.001$ (two-way ANOVA with Sidak's multiple comparisons test). **c** P-p44/42 MAPK levels do not substantially change upon treatment of LLC, 4T1, and Saos-2 metastatic cancer cells with rFibcd1 (10 and 100 ng/mL for 30 and 120 min), apart for Saos-2 cells treated with the highest rFibcd1 dose. **d** rFibcd1 induces limited/minimal changes in P-p44/42 MAPK in most osteosarcoma, colorectal adenocarcinoma, melanoma, and breast cancer cell lines. However, rFibcd1 induces P-p44/42 in 67NR and 143B-Luc cells. See also Supplementary Fig. 3b. **e** rFibcd1 primarily consists of a fibrinogen-related domain (Fig. 2a). Combinations of a and b integrins are known to act as receptors for fibrinogen, including a2b-b3 integrins. **f** RNA-seq and qRT-PCR indicate that integrin a2b is expressed in C2C12 myotubes, which respond to rFibcd1 by increasing P-p44/42 levels but is not expressed in cancer cells that do not readily respond to rFibcd1 (LLC, 4T1, E0771, Ep5). 67NR cancer cells that respond to rFibcd1 (as indicated by higher P-p44/42; Fig. 3d and Supplementary Fig. 3b) display *Itga2b* mRNA levels similar to those of C2C12 myotubes, suggesting that Itga2b might be a receptor for rFibcd1; $n = 2$ (RNA preps from independent cell cultures). **g** Itga2b siRNAs impede the rFibcd1-mediated increase in P-ERK, indicating that Itga2b is needed for the optimal response of muscle cells to rFibcd1. The quantitation of P-ERK levels relative to tubulin is shown in **a**, **c**, **g**. Source data are provided in the Source Data file.

for inflammatory cachectic cytokines (Supplementary Data 2). Together, these findings indicate that cachectic factors produced by LLC cells, such as IL-6, LIF, and TNF-α[96,102], induce inflammatory signaling in the diaphragm muscle and that intraperitoneal injection of rFibcd1 can impede such transcriptional changes (Fig. 4l).

Previous studies have shown that muscle proteolysis is a key mechanism responsible for myofiber atrophy induced by cancer[12,24]. In this respect, further analysis revealed that LLC cancer cells increase the expression of cathepsins in the diaphragm muscle, such as *Ctss*, *Ctse*, *Ctsw*, *Ctsz*, *Ctsc*, *Ctsh*, and *Ctso* (Fig. 4m and Supplementary Data 2). Because cathepsin are lysosomal proteolytic enzymes that have been previously implicated in myofiber atrophy[103–106], they likely contribute to the proteolysis responsible for LLC cancer-induced muscle mass loss. However, rFibcd1 largely rescued the upregulation of cathepsin expression due to LLC cancers (Fig. 4m and Supplementary Data 2). Other proteolytic enzymes were also similarly regulated by LLC cancers and rFibcd1, such as *Napsa*, a member of the peptidase A1 family of aspartic proteases (Fig. 4m). Moreover, *ubiquitin D* (*Ubd*) expression was induced by LLC cancers but repressed by rFibcd1 (Supplementary Data 2) and may sustain proteolysis via the ubiquitin-proteasome system. Lastly, ubiquitin ligases were also regulated and are known to have important roles in muscle proteolysis during wasting[24,103,107,108]. Interestingly, some of them were upregulated in the diaphragm of mice with LLC cancers but this was rescued by rFibcd1 injection. These included *Hgs* and *Syvn1* (Supplementary Data 2), which were previously shown to negatively regulate myofiber size[60]. However, LLC cancers did not significantly induce other ubiquitin ligases involved in myofiber atrophy[24,103,107,108], such as *Fbxo32/Atrogin-1* and *Trim63/MuRF1* (Supplementary Data 2). Although relatively few genes were downregulated in response to LLC cancers, there were some that have been previously associated with myofiber hypertrophy, such as *Pparg1a/PGC1a*[109], which was downregulated in LLC but not in LLC + rFibcd1 versus control (Fig. 4m and Supplementary Data 2).

Altogether, these findings indicate that rFibcd1 hinders some of the molecular changes associated with cancer cachexia in the diaphragm, including expression of genes already implicated in myofiber atrophy via their roles in inflammation, proteolysis, and other processes.

**A patient-derived melanoma xenograft induces progressive atrophy of diaphragm myofibers.** LLC cancer-induced cachexia is an established disease model that has led to many insights into the mechanisms of muscle wasting in cancer[60,96,97]. However,

there is growing appreciation that patient-derived orthotopic cancer xenografts provide cachexia models that more closely mimic disease progression in humans[97,110]. To this purpose, we have identified a cachexia-inducing ("cachectic") orthotopic patient-derived melanoma xenograft from the Childhood Solid Tumor Network collection established at St. Jude Children's Research Hospital[111–115]. Specifically, we have established a model of cancer cachexia induced by a patient-derived melanoma xenograft (MAST360B/SJMEL030083_X2) and compared it to mice bearing a control melanoma xenograft that does not induce cachexia (MAST552A/SJMEL031086_X1; "non-cachectic").

Compared to the commonly used LLC model of cancer-induced cachexia (Fig. 4), the melanoma model (MAST360B) used here induces severe muscle mass loss, i.e. ~25% versus ~10% loss found with LLC (Supplementary Fig. 5a). The body weight loss induced by this cachectic melanoma xenograft is characterized by a decline in the mass of the white adipose tissue, liver, pancreas, ovaries, heart, and skeletal muscles (i.e. soleus, tibialis anterior, gastrocnemius, diaphragm), (Supplementary Fig. 5b).

To better define how cachexia induced by this melanoma xenograft impacts the diaphragm, we have employed histological and transcriptional analyses to follow the progressive development of myofiber atrophy in the diaphragm muscle. There was no significant change in myofiber size at 2 weeks (Fig. 5a, d) and 4 weeks (Fig. 5e–h) post tumor implantation. However, myofiber atrophy was observed at week 6 (Fig. 5i–l) and further developed by week 8 from tumor injection (Fig. 5m–p).

Analysis of gene expression changes indicates that diaphragm atrophy associates with a set of 1033 genes (Fig. 5q) and that upregulated gene categories include secreted/immune proteins and inflammatory pathways (Fig. 5r), whereas relatively few genes were downregulated (Fig. 5s). Similar transcriptional changes were also found to occur in the tibialis anterior and soleus muscles (Supplementary Fig. 5c, d and Supplementary Data 4).

Altogether, these studies identify myofiber atrophy and transcriptional upregulation of inflammatory genes as components of diaphragm cachexia induced by pediatric melanoma xenografts.

**rFibcd1 partially rescues diaphragm myofiber atrophy induced by a patient-derived melanoma xenograft.** On this basis, we next tested whether rFibcd1 impedes diaphragm myofiber atrophy induced by a cachectic melanoma xenograft (MAST360B/SJMEL030083_X2), compared to a control melanoma that does not induce muscle wasting (MAST552A/SJMEL031086_X1).

For these experiments, ~$10^6$ pediatric melanoma xenograft cells were injected subcutaneously in the flank and allowed to grow for ~7 weeks. To test the efficacy of rFibcd1, 3 intraperitoneal

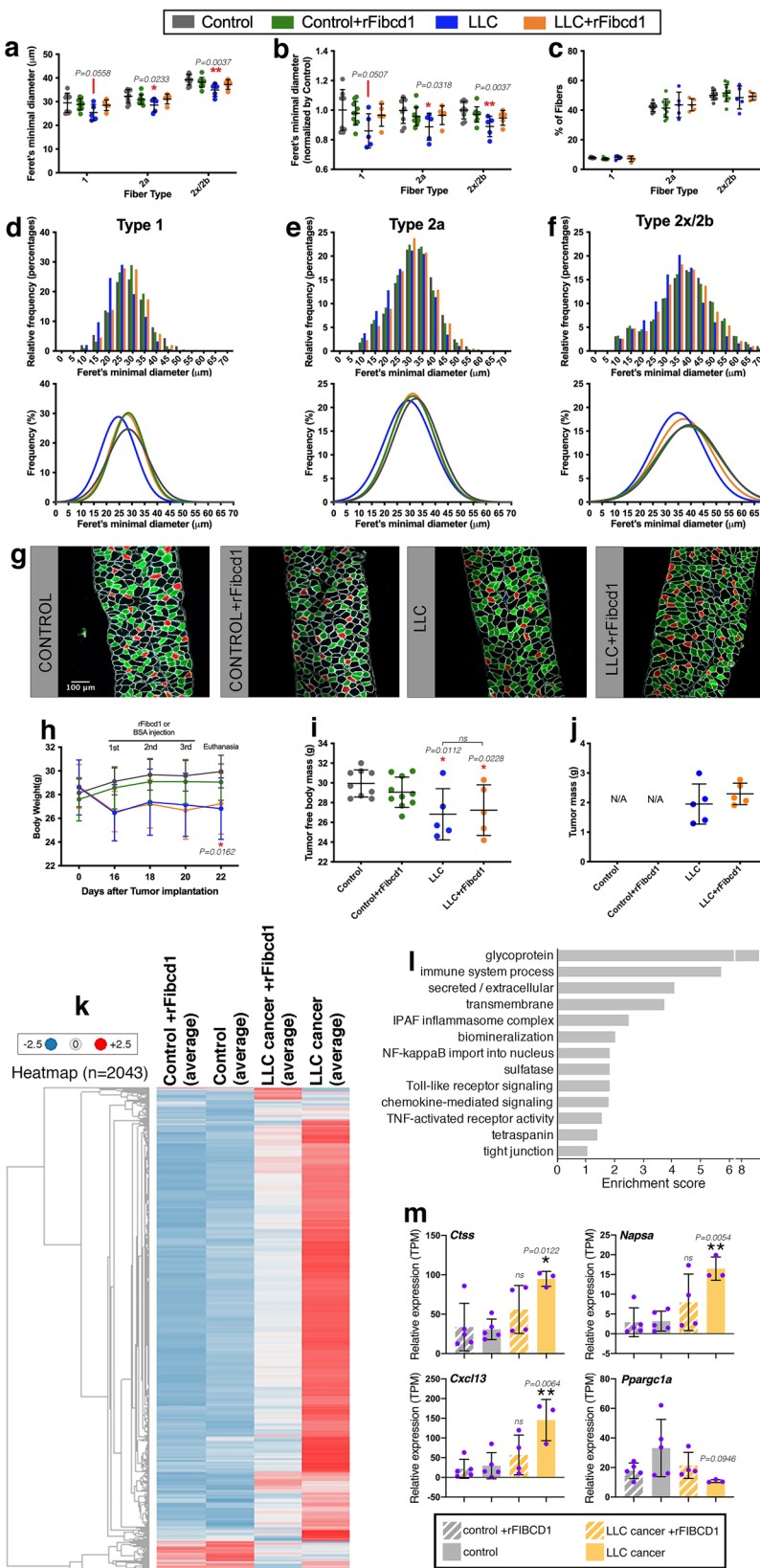

injections (every other day, with 3 mg/kg of rFibcd1 in PBS with 1% BSA) were done in the last week of xenograft growth before euthanasia in half of the mice carrying xenografts whereas the other half received mock injections (PBS with 1% BSA).

By examining the size of myofibers in the diaphragm muscle of these mice, we found that the cachectic melanoma induces significant atrophy, as indicated by the average Feret's minimal diameter, compared to mice injected with a control non-cachectic

**Fig. 4 rFibcd1 rescues LLC cancer-induced myofiber atrophy of the diaphragm. a–f** Myofiber size of diaphragms from control mice treated with mock ($n = 9$) or with rFibcd1 ($n = 10$), and from mice that carry LLC tumors and that are treated with mock ($n = 5$) or with rFibcd1 ($n = 5$). Treatment consisted of 3 intraperitoneal injections of rFibcd1 (1 mg/Kg) or mock injection of 1% BSA (vehicle). **a** Feret's minimal diameter of type 1, 2a, and 2x/2b myofibers. **b** Feret's minimal diameter normalized to controls. rFibcd1 partially rescues cancer-induced myofiber atrophy. **c** Percentage of type 1, 2a, and 2x/2b myofibers. In (**a–c**), mean ± SD are shown and analyzed with two-way ANOVA with Sidak's multiple comparisons. **d–f** Frequency and gaussian distribution of Feret's minimal diameters for type 1 **d**, type 2a **e**, and type 2x/2b myofibers **f**. **g** Representative images of diaphragm strips immunostained for type 1 (red), type 2a (green), and type 2x/2b myofibers (black). Laminin (white) delineates all myofibers, and DAPI (blue) the nuclei. **h-i**, Body weight and tumor-free body mass decrease in response to LLC tumor growth. **i–j** Intraperitoneal injection of rFibcd1 does not affect the tumor-free body mass and the tumor mass, indicating that rFibcd1 exerts local rather than system effects. N/A, not applicable. Data are mean ± SD. *$P < 0.05$ (two-way ANOVA with Sidak's multiple comparisons test). The n is the same as reported in (**a**) and refers to mice in **h-i**, and to the mass of tumors in **j**. **k** Heatmap of 2043 differentially-expressed genes in diaphragms from control mice treated with mock ($n = 5$) or with rFibcd1 ($n = 5$), and mice that carry LLC tumors and are treated with mock ($n = 3$) or with rFibcd1 ($n = 4$). **l** GO term analysis of genes (Supplementary Data 2) that are upregulated (230) and downregulated (17) ($P < 0.05$ and Log2R < -0.5 and Log2R > 0.5) in the diaphragm of mice with LLC but not if treated with rFibcd1. **m** These include genes related to inflammation (*Cxcl13*), proteolysis (*Ctss* and *Napsa*) and metabolism (*Ppargc1a*). Data are means ± SD with $n$ indicated; *$P < 0.05$ and **$P < 0.01$ (two-way ANOVA with Tukey's multiple comparisons). Source data are provided in the Source Data file.

melanoma. Importantly, all myofiber types (1, 2a, 2x/2b) were affected, indicating the cachectic factors released by this melanoma have the general capacity to induce wasting (Fig. 6a). However, rFibcd1 injection significantly reduced the atrophy induced by cancer cachexia in type 1 and 2a myofibers, whereas smaller effects were seen for type 2b myofibers (Fig. 6a). Similar results were found by examining the average Feret's minimal diameter values normalized to control (Fig. 6b). These changes in myofiber size were not due to a variation in the percentage of myofiber types present in the diaphragm (Fig. 6c).

On this basis, we next examined the relative frequencies of myofibers in the diaphragm muscle. Analysis of type 1 (Fig. 6d), type 2a (Fig. 6e), and type 2x/2b (Fig. 6f) revealed that the cachectic melanoma shifts the distribution of myofiber sizes towards lower values compared to a control non-cachectic melanoma (Fig. 6g). However, rFibcd1 partially reinstated myofiber sizes, in particular for type 2a myofibers (Fig. 6a–g). Considering that type 2a myofibers are the most abundant in the diaphragm muscle (~60% of all myofibers, Fig. 6c) and are also abundant (~50%) in the diaphragm in humans[116,117], these findings suggest that rFibcd1 may provide a significant means for preserving myofiber size in the diaphragm of cancer patients.

Further analyses revealed that the anti-cachectic effects of intraperitoneal injections of rFibcd1 on the diaphragm muscle were not due to a systemic decrease in cancer growth and body wasting but were rather due to local effects on the diaphragm muscle (Fig. 6h–j). Specifically, body weight significantly decreased in mice that carried the MAST360B/SJMEL030083_X2 cachectic melanoma compared to mice injected with the MAST552A/SJMEL031086_X1 non-cachectic melanoma (Fig. 6h). However, body weight equally declined in mice that carried the MAST360B/SJMEL030083_X2 cachectic melanoma and that received either rFibcd1 or mock intraperitoneal injections (Fig. 6h). The same conclusions were reached also via the analysis of the tumor-free body mass (Fig. 6i). Similarly, intraperitoneal injection of rFibcd1 does not significantly impact the weight of the tibialis anterior, a hindlimb muscle (Supplementary Fig. 6). Moreover, intraperitoneal rFibcd1 injection did not impact tumor burden (Fig. 6j). Altogether, these findings indicate that reduction in cancer-induced atrophy of diaphragm myofibers is not due to systemic effects of intraperitoneal injection of rFibcd1 on body wasting and tumor growth. Lastly, circulating inflammatory cytokines were not significantly modulated by rFibcd1 (Fig. 6k and Supplementary Fig. 7), suggesting that the capacity of rFibcd1 to decrease cancer-induced atrophy of diaphragm myofibers primarily results from its action on muscle cells.

**rFibcd1 rescues muscle transcriptional changes induced by cachectic melanomas.** Because rFibcd1 rescues myofiber atrophy induced by patient-derived xenografts of cachectic melanomas (Fig. 6), we next examined whether rFibcd1 rescues the muscle transcriptional changes associated with myofiber atrophy in this model. RNA-seq revealed that rFibcd1 regulates the expression of several genes, some of which are modulated also in cachectic versus non-cachectic melanomas. Altogether, as observed for transcriptional changes due to LLC cancers (Fig. 5), treatment with rFibcd1 reduces the magnitude of gene expression changes induced in the diaphragm muscle in response to cachectic versus non-cachectic melanomas (Fig. 7a).

To further examine the impact of rFibcd1 on muscle cachectic changes, we examined the GO term categories that are enriched in genes that are significantly regulated in the muscle of mice with cachectic melanomas but not in the muscle of mice with cachectic melanomas upon rFibcd1 treatment (Fig. 7b and Supplementary Data 5). Examples of genes upregulated by rFibcd1 treatment (Fig. 7c) include *Capza1*, a capping protein necessary for the maintenance of thin filaments in vertebrate skeletal muscle[118], and troponin *Tnni1*, which may help preserve skeletal muscle contractile functions[119]. Expression of *Arrdc2*, a gene previously implicated in muscle atrophy[120], increases in response to cachectic melanoma but this is partially prevented by rFibcd1. Likewise, rFibcd1 impedes the cachexia-mediated upregulation of: *Ddit4/REDD1*, a repressor of mTOR signaling and mediator of muscle wasting[121–123]; the secretoglobin *Scgb3a1*, which is induced by cachexia[124]; the skeletal-muscle specific kelch-like protein *Klhl38*, a substrate-specific adapter for cullin E3 ubiquitination implicated in muscle wasting[125,126]; the transcription factor *Klf15*, a direct target of glucocorticoid signaling that like Ddit4/REDD1 inhibits mTOR signaling and induces muscle wasting[123,127,128]; and the transcription factor *Zbtb16*, which is induced in brown adipose tissue and skeletal muscle during thermogenesis[129], which is stimulated by cachexia[130–132]. Together, these findings indicate that intraperitoneal injection of rFibcd1 can impede some of the transcriptional changes associated with melanoma-induced cachexia (Fig. 7).

Because of its key role in respiration, diaphragm function is an important determinant of overall exercise capacity[133–135]. On this basis, we next used treadmill tests[136,137] to determine whether i.p. injection of rFibcd1 targeting the diaphragm improves motor function in mice with cachexia. To this purpose, we compared mice with either cachectic or non-cachectic orthotopic melanoma xenografts, and which received 3 i.p. injections of either rFibcd1 or mock (BSA) a week before testing. The total running time, the maximum and average speed, and the total distance run were

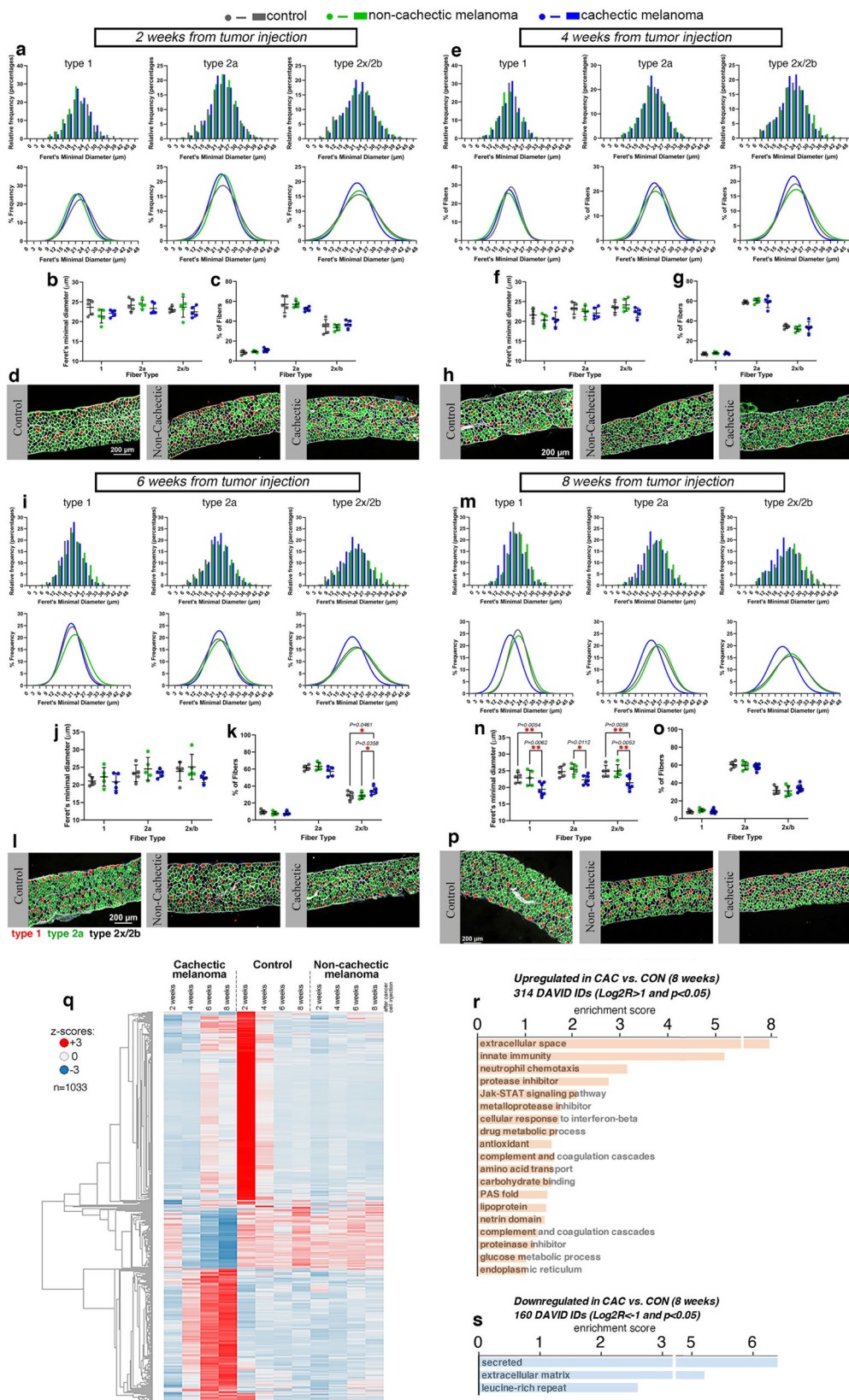

lower in mice implanted with cachectic versus non-cachectic melanoma xenografts (Fig. 7d). The treadmill capacity of rFibcd1-treated cachectic mice trended towards improvement compared to that of mock-treated cachectic mice but this difference did not reach statistical significance. However, cachectic mice treated with rFibcd1 have a treadmill capacity that is also not statistically different from that of non-cachectic mice (Fig. 7d), presumably because of the capacity of rFibcd1 to attenuate cancer-induced transcriptional changes and myofiber atrophy in the diaphragm (Figs. 6–7).

**Fig. 5 Cachexia caused by a pediatric melanoma xenograft induces myofiber atrophy in the diaphragm. a–p** Analysis of myofiber size in diaphragm muscles of mice undergoing wasting at 2 **a–d**, 4 **e–h**, 6 **i–l**, and 8 **m–p** weeks post implantation of a cachexia-inducing ("cachectic") melanoma xenograft, compared to a cachexia-non-inducing ("non-cachectic") melanoma xenograft or mock injection of PBS (*n* = 5/group). At the early stages of cachexia progression (2-4 weeks), there is no substantial decline in myofiber size, as indicated by the analysis of the Feret's minimal diameter **a–b**, **e–f**. At later stages (6–8 weeks), there is a trend towards a decline in the size of type 1, 2a, and 2x/2b myofibers at 6 weeks of age **i**, **j**, **l**, which however is significant only after 8 weeks from cancer cell injection **m**, **n**, **p**. There are no changes in the relative proportion of myofiber types found in diaphragm muscles **c**, **g**, **o**, apart for a slight increase in type 2x/2b myofibers at 6 weeks (**k**). Mean values ± SD are shown in panels **b–c**, **f–g**, **j–k**, and **n–o**. Statistical analysis was done by using two-way ANOVA with Sidak's multiple comparisons test. **q** Heatmap of 1033 genes that are most highly modulated by melanoma-induced cachexia in the diaphragm, compared to controls. **r** Upregulated genes include secreted proteins, proteins involved in innate immunity and neutrophil chemotaxis, and metalloprotease inhibitors. **s** Downregulated genes are also enriched for secreted and extracellular matrix proteins. Genes modulated with $P < 0.05$ and Log2R > 1 **r** and Log2R < -1 **s** in cachectic versus control at 8 weeks post tumor implantation are shown. Source data are provided in the Source Data file.

## Discussion

Over the past decades, many studies have deciphered the signaling mechanisms responsible for skeletal muscle wasting, thereby highlighting several potential therapeutic targets[12,24,138]. However, despite increased understanding of the mechanisms involved and extensive industry efforts, it has proven difficult to develop interventions for muscle atrophy, for which there is an unmet medical need.

A promising avenue for translational applications in this field is provided by myokines, some of which are endogenous regulators of myofiber size[36,40,51–53,57,58]. In this study, we have identified evolutionary-conserved myokines that regulate myofiber size (Fig. 1). In particular, we have studied the mechanism of action and potential therapeutic use of one of them, Fibcd1. Similar to another myokine that preserves myofiber size, Irisin/FNDC5[40,139], secreted Fibcd1 is generated by the proteolytic cleavage of a transmembrane version (Fig. 2) and is similarly detected in the human plasma (Supplementary Fig. 8). Local injection of recombinant secreted Fibcd1 attenuates myofiber atrophy induced by cancer cachexia in the diaphragm muscle (Figs. 2–8). On this basis, recombinant Fibcd1 (rFibcd1) may be used as a therapeutic agent to impede or to improve recovery from muscle wasting. For example, intraperitoneal injection into the abdominal cavity[140] may provide an effective administration route for targeting the wasting of the diaphragm, a key muscle which is necessary for respiration and that is located between the abdominal and thoracic cavities. Such intraperitoneal application of rFibcd1 to preserve the size and function of diaphragm myofibers could improve the prognosis of patients with cancer cachexia given that wasting of the diaphragm is a key determinant of mortality in these patients[11,98].

We have found that rFibcd1 preserves myofiber size via ERK signaling (Fig. 2), consistent with previous studies that have demonstrated a key role for ERK in promoting protein synthesis and in maintaining skeletal muscle mass and function in mice[61–64]. Interestingly, another secreted factor (FGF19) has also been previously reported to preserve myofiber size via ERK signaling[58]. However, because ERK also promotes cell proliferation[87], a possible side effect of rFibcd1 treatments may consist in the promotion of tumor growth. To test this hypothesis, we have examined the response of cancer cells to rFibcd1 and found that rFibcd1 does not induce ERK activity in many cancer cells, different from myotubes (Fig. 3). This selectivity for muscle cells depends at least in part on expression in muscle cells of *Itgα2b* integrin, which is needed for the optimal response to rFibcd1 (Fig. 3), whereas *Itgα2b* is not expressed by many cancers (Supplementary Fig. 4). However, a cancer cell line displayed high *Itgα2b* expression and higher P-Erk levels in response to rFibcd1 (Fig. 3), suggesting that *Itgα2b* and/or alternative receptors may also transduce rFibcd1 signaling in certain cancer cells. Nonetheless, intraperitoneal injections of rFibcd1 did not increase

tumor growth in vivo, as estimated in experiments with melanoma and LLC cancers (Figs. 4, 6). Together, these studies suggest that certain cancers may not respond to rFibcd1, and that intraperitoneal rFibcd1 may prove to be a safe therapeutic in such contexts, given its capacity to promote ERK signaling in diaphragm myofibers but not in cancer cells. These findings appear to be of particular interest considering that the cancer cell lines and tumor types here used are highly metastatic to the lungs[89–92] and that metastatic capacity is strongly associated with the induction of muscle wasting[10,141]. On this basis, rFibcd1-based treatments may benefit cancer patients that develop respiratory dysfunction due to lung metastases and to concurrent cancer-induced wasting of the diaphragm.

Similar to Fibcd1, our findings and previous studies indicate that ERK signaling in muscle cells is necessary to preserve myofiber size in mice. Specifically, ERK signaling is necessary for the maintenance of skeletal muscle mass[63] and its inhibition prevents IGF-I-induced myofiber hypertrophy[64]. Moreover, ERK has been shown to be necessary to maintain adult myofiber size and their neuromuscular junctions in mice[58,61,62]. Consistently, our screen for myokines that regulate myofiber growth in *Drosophila* (Fig. 1) has found that muscle-restricted interventions that are known to promote ERK signaling induce myofiber hypertrophy (Fig. 1f, g). For example, overexpression of *Pvf1* (homologous to PDGF/VEGF) induces myofiber hypertrophy, whereas overexpression of *argos*, an antagonist of EGF Receptor signaling, induces atrophy (Fig. 1f, g).

In addition to being necessary for the maintenance of myofiber size[58,61,62], chronic activation of Erk signaling in skeletal muscle was found to induce a switch from fast-twitch to slow-twitch myofibers, which resulted in enhanced metabolic capacity and fatigue resistance of skeletal muscles and reduced the severity of muscular dystrophy in mice[142]. Consistent with these muscle protective functions of Erk signaling, its phosphorylation status (i.e. activity) was found to increase in myofibers upon exercise[143–146]. However, Erk phosphorylation was also reported to increase during cachexia in muscle[147]. Although the significance of this contrasting regulation is unclear, Erk activity might be induced in both conditions in response to cytokine signaling, and act as a compensatory mechanism to cope with the stress and tissue damage that occurs with both exercise[148] and cachexia[149]. On this basis, Erk induction might contribute to an adaptive response to maintain muscle homeostasis.

This and previous studies reporting a protective role of Erk signaling in preserving myofiber size are in apparent contradiction with other studies that have found that systemic administration of MEK/ERK inhibitors in cachectic mice can cure muscle wasting[150]. However, such anti-wasting effect of MEK/ERK inhibitors seem to stem primarily from inhibition of ERK signaling in cancer cells rather than muscle[105,106]. Specifically, MEK/ERK inhibitors were found to reduce the expression of

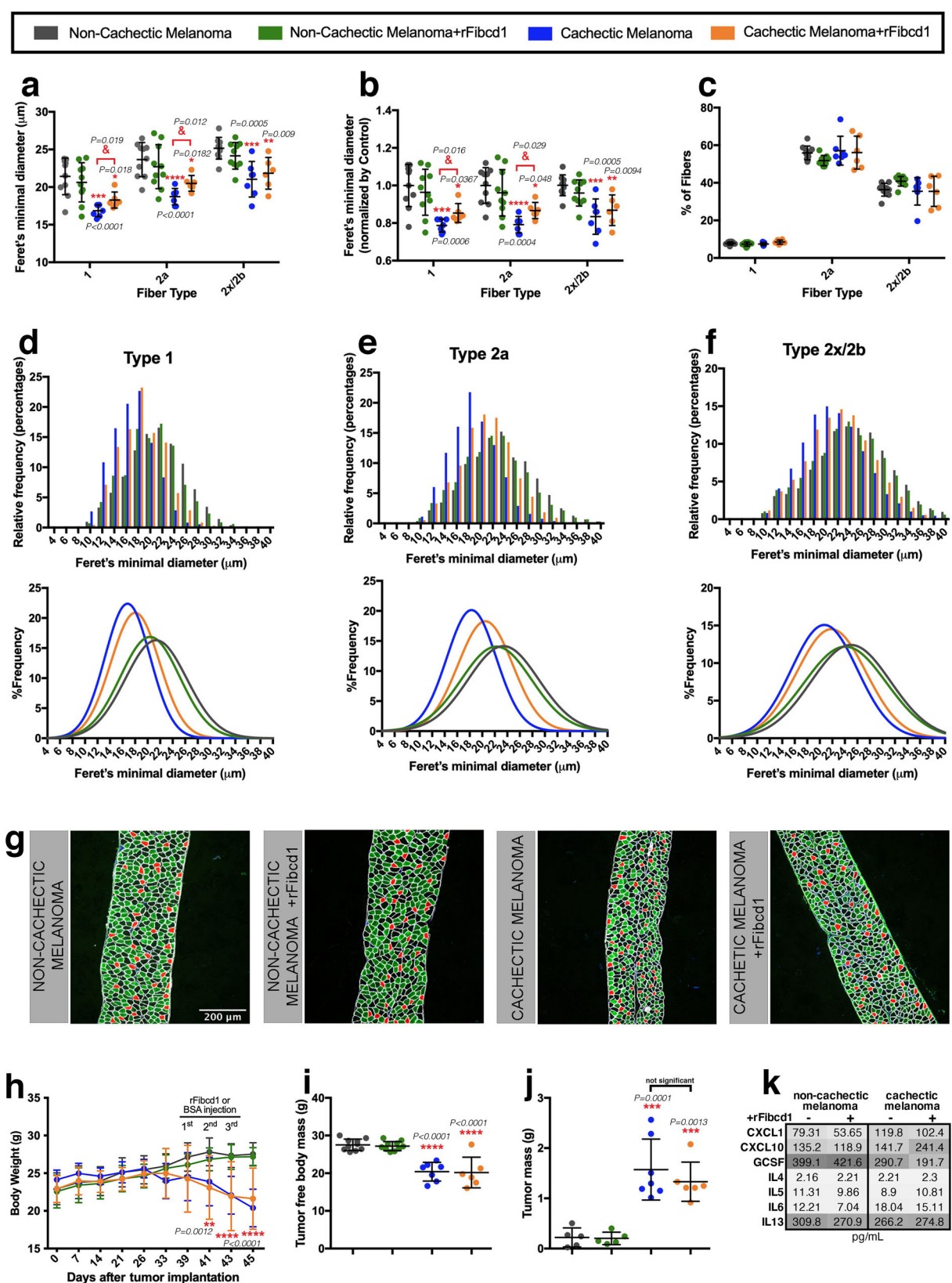

cachectic cytokines produced by cancer cells[105,106], independently from reduction in cancer growth, which can also occur in response to administration of MEK/ERK inhibitors[105,106]. In particular, MEK/ERK inhibitors were found to reduce the circulating levels of IL-6[150,151], which is a cachectic cytokine produced by cancer cells and key for the induction of muscle wasting[114,115].

Therefore, inhibition of ERK signaling in cancer cells likely reduces muscle wasting by reducing cancer growth and/or cachectic cytokine production[114,115,152].

On the other hand, ERK signaling in skeletal muscle is known to limit myofiber atrophy via the capacity of ERK to phosphorylate and inactivate the atrophy-inducing kinase GSK3β[81,82], and

**Fig. 6 rFibcd1 rescues diaphragm myofiber atrophy induced by a patient-derived melanoma xenograft. a–f** Myofiber size from diaphragms of control mice with a non-cachectic melanoma and treated with mock (n = 10) or with rFibcd1 (n = 10), and from mice that carry a cachectic melanoma and are treated with mock (n = 7) or with rFibcd1 (n = 6). Treatment with rFibcd1 consisted of 3 intraperitoneal injections of rFibcd1 (3 mg/Kg) or of the vehicle (1% BSA). **a** Feret's minimal diameter of type 1, 2a, and 2x/2b myofibers. **b** Feret's minimal diameter normalized to controls. **c** Percentage of type 1, 2a, and 2x/2b myofibers. In **a–c**, mean values ± SD are shown, and were analyzed with two-way ANOVA with Sidak's multiple comparisons and with two-tailed unpaired t-tests. **d–f** Frequency and gaussian distribution of Feret's minimal diameters for type 1 **d**, type 2a **e**, and type 2x/2b myofibers **f**. **g** Representative micrographs of diaphragm strips immunostained for type 1 (red), type 2a (green), and type 2x/2b myofibers (black). Laminin (white) delineates myofibers, and DAPI (blue) the nuclei. **h–i** Body weight and tumor-free body mass progressively decline in response to growth of a cachectic melanoma but not in response to a control non-cachectic melanoma. **h–j**, Intraperitoneal injection of rFibcd1 does not affect the body weight, tumor-free body mass, and the tumor mass, indicating that rFibcd1 does not rescue myofiber atrophy by impacting tumor growth and general body wasting. In **h–j** data are mean ± SD; ***P < 0.0001 compared to control non-cachectic melanoma group; &P < 0.05 compared to the control cachectic melanoma group. Statistical analysis was done by using two-way ANOVA with Tukey's multiple comparisons test and two-tailed unpaired t-test. See also Supplementary Fig. 5. **k** Heatmap showing the levels of cytokines (pg/mL) from the plasma of mice from the groups described above. Lack of Fibcd1 treatment is indicated by a (−), whereas (+) denotes rFibcd1 administration. Color key indicates low plasma levels (light gray) and high plasma levels (high gray) of cytokines. See also Supplementary Fig. 7. Source data are provided in the Source Data file.

via ERK-mediated activation of MNK1/2-eIF4E and the consequent promotion of protein synthesis[81,83,84]. Moreover, in addition to general roles of ERK signaling in preserving muscle mass and function[58,61–64], ERK signaling has been found to promote protein synthesis and preserve the function of the diaphragm[81,153], indicating that ERK signaling is indeed protective for this key skeletal muscle. Altogether, rFibcd1-mediated promotion of ERK signaling in the diaphragm muscle without concomitant induction of ERK in cancer cells appears as an appropriate strategy to contrast myofiber atrophy.

Interestingly, rFibcd1 also activates ERK in control conditions but this does not lead to myotube hypertrophy. Possibly, there are limitations (independent from Erk signaling) that impede additional growth in normal conditions. In agreement with our findings, previous studies have shown that Erk signaling is primarily needed for the maintenance of myofiber size rather than for the induction of myofiber hypertrophy[58,61–64].

In addition to modulating ERK signaling (Fig. 3), we have found that rFibcd1 reduces the expression of inflammatory genes (Fig. 4) and myofiber atrophy (Figs. 4 and 6) in the diaphragm skeletal muscle. Consistent with a role of Fibcd1 in reducing tissue inflammation, recent studies have found that gut-specific Fibcd1 overexpression reduces intestinal inflammation[76,154] and impedes weight loss induced by chemotherapy in mice[154]. Taken together, our study and these recent publications suggest that Fibcd1 reduces tissue inflammation, which is a major driver of cancer-induced muscle wasting[24]. However, paradoxically, high FIBCD1 expression in cancer cells has been found to correlate with poor prognosis and survival in patients with gastric and liver cancer[155,156]. A possible explanation for this dichotomy is that FIBCD1 expression by cancer cells impedes local inflammatory responses and hence allows cancer immune escape, which is a key mechanism of cancer progression and metastatic dissemination[157].

Although many studies have investigated the mechanisms responsible for cachexia in the context of cancer types that are prevalent in adults, relatively little is known on the mechanisms and role of cachexia in pediatric cancer patients[158]. In this study, we have established a model of cancer cachexia induced by a patient-derived orthotopic pediatric melanoma xenograft. We propose that this model may provide a suitable setting for testing therapeutic interventions and to extend the portfolio of tools available for basic science studies on cachexia.

In addition to cancer cachexia, muscle wasting is a common issue that occurs with aging and other common diseases such as diabetes, kidney failure, neuromuscular disorders, and infections[24,159]. In particular, inflammatory conditions such as chronic obstructive pulmonary disease (COPD) and lung infections induce muscle atrophy and are lethal because of the

synergistic decline in the function of the lung tissue and wasting of the diaphragm[160,161], which also occurs during normal aging[162–164]. Therefore, treatment with rFibcd1 may help preserve pulmonary function during aging and in the course of other illnesses that compromise diaphragm function[165], in conjunction with inspiratory muscle training[19–23] and yet-to-develop pharmaceutical treatments.

In summary, our study indicates that Fibcd1 may help reinstate myofiber size in the diaphragm muscle (Fig. 8), which is a key muscle impacted by cancer cachexia.

## Methods

***Drosophila* larval screening and body wall skeletal muscle analysis.** Flies were maintained at 25 °C with 60% humidity in tubes containing cornmeal/soy flour/yeast fly food. Fly stocks were obtained from either the Bloomington *Drosophila* Stock Center (BDSC), The National Institute of Genetics Fly Stocks (NIG-Fly), or the Vienna *Drosophila* Resource Center (VDRC) and are listed in Supplementary Data 1 alongside the full results of larval screening for regulators of myofiber size.

To define the list of *Drosophila* stocks for transgenic RNAi and overexpression of evolutionary conserved myokines, the following procedures were used: first, a list of 788 predicted secreted factors (based on the GLAD, Gene List Annotation for *Drosophila*[166]) was filtered to retain only 274 *Drosophila* secreted factors with sufficient homology to humans, i.e. with a score ≥2 as defined by DIOPT[167]; second, the list was further filtered to retain only 111 *Drosophila* secreted factors with sufficient skeletal muscle expression, as defined by RNA-seq FPKM ≥ 4; lastly, the function of these evolutionary-conserved muscle-secreted factors was screened with 508 stocks for RNAi and overexpression.

For the screen, we used similar procedures as we recently used[60,66]. Specifically, virgin *Mef2-Gal4* females (10 flies for each cross) were mated with males (5 flies for each cross) of each RNAi or overexpression stock and transferred to new food tubes every 3 days to avoid overcrowding. Breeders were maintained at 25 °C. The sizes of 3rd instar larvae were assessed qualitatively as either bigger or smaller than control larvae. As secondary validation, skeletal muscles from selected candidates were analyzed by dissecting larvae into filets and by exposing the ventral lateral muscles VL3 and VL4 (each consisting of a single myofiber) from abdominal segments 2-4, which were then fixed for 30 min with 4% EM-grade PFA in PBS without Ca²⁺ and Mg²⁺ [59]. Following washes, larval body wall muscles (typically from 10 larvae) were stained with DAPI (1:1000) and imaged to detect the endogenous fluorescence of a Mhc-GFP fusion protein. Subsequently, the total area, width, and length of VL3 + VL4 myofibers was quantified with the Zeiss Zen software. All schemes were drawn with Biorender.

**Production of recombinant mouse Fibcd1.** The ~38-kDa C-terminal part of mouse Fibcd1 (consisting of 282 amino acids, approximately corresponding to the secreted Fibcd1 fragment retrieved from the cell culture medium, as shown in Fig. 2) was cloned into the pAcGP67-B vector with *BamH*I and *Bgl*II. Expression of the C-terminal part of mouse Fibcd1 was achieved with the Bac-to-Bac Baculovirus Expression System (Invitrogen) and the recombinant protein (rFibcd1) retrieved in PBS (GIBCO, #10010023) with 1% BSA. The identity of purified rFibcd1 was confirmed by mass-spectrometry.

**C2C12 culture, myotube siRNA transfection, and rFibcd1 treatment.** C2C12 mouse muscle cells (ATCC, #CRL-1772) were cultured at 37 °C with 5% CO₂ in DMEM (high glucose DMEM, with Glutamax, GIBCO, #10566016) containing 10% fetal bovine serum (GIBCO, #10437-028), and 1% penicillin/streptomycin (10,000 U/

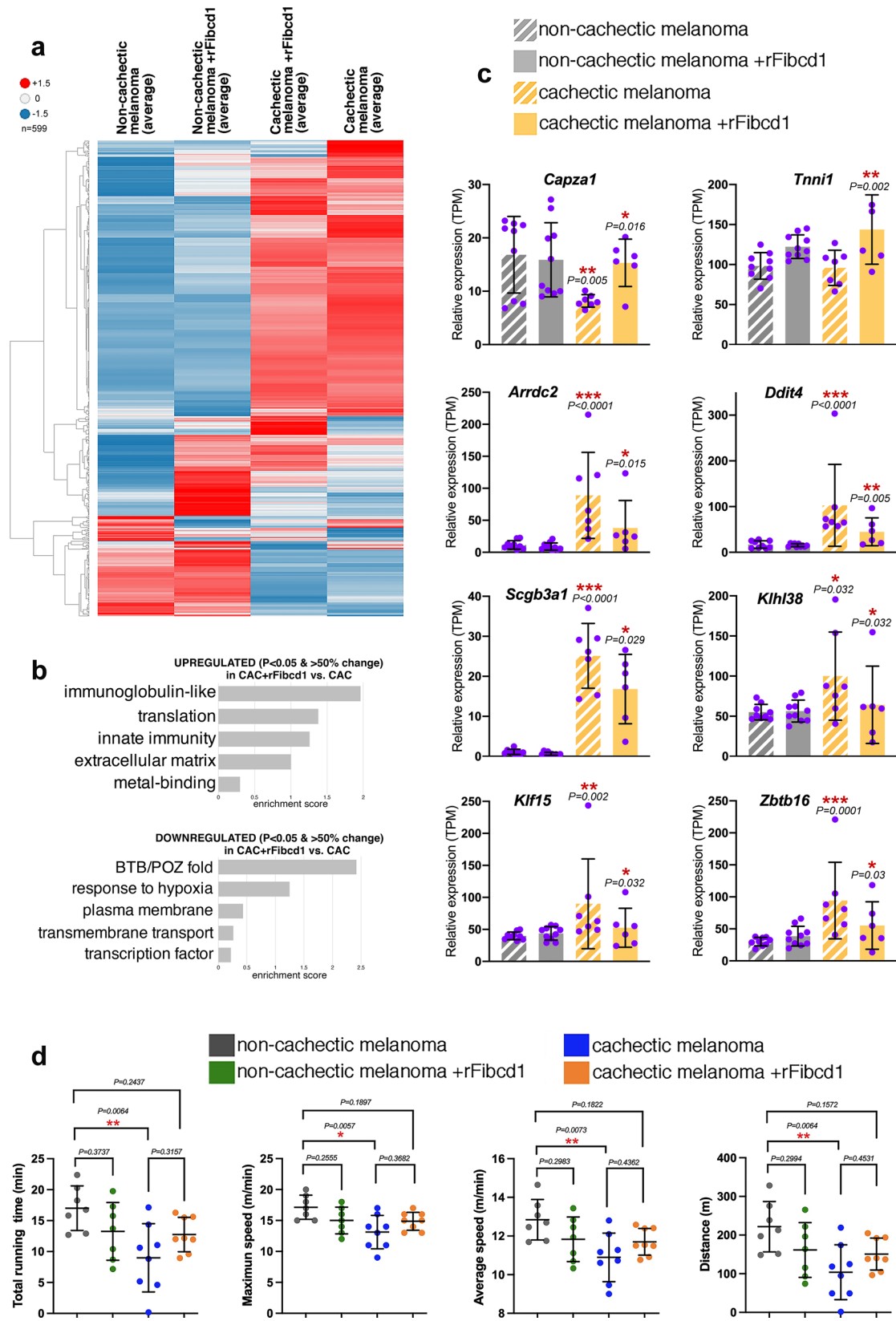

mL, GIBCO, #15140122). C2C12 cells were maintained as myoblasts with media containing 10% fetal bovine serum and switched to 2% horse serum-containing media (GIBCO, #26050070) to induce differentiation into myotubes when near confluence. 4 days after differentiation, myotube-enriched cultures were generated by adding media containing 4 mg/mL of Cytosine b-D-arabinofuranoside (Ara-C,

Sigma, #C1768) for a further 2 days at which point the remaining myotubes were transfected with siRNAs. To this purpose, myotubes were transfected with 50 mM siRNAs targeting the specified gene or with control non-targeting (NT) siRNAs, by using a ratio of 2 mL Lipofectamine 2000 (Invitrogen, #11668027) to 50 pmol of siRNA in OptiMEM (GIBCO, #31985062), as previously done[60]. Myotube size was

**Fig. 7 rFibcd1 reduces transcriptional and functional changes induced by cachectic melanomas. a** Heatmap of 599 genes that are differentially expressed in diaphragm muscles of mice bearing non-cachectic melanomas and either mock-treated ($n = 9$) or treated with rFibcd1 ($n = 10$), and mice that carry a cachectic melanoma and that are either mock-treated ($n = 7$) or treated with rFibcd1 ($n = 6$). rFibcd1 represses some of the gene expression changes that characterize diaphragms from mice implanted with cachectic versus the non-cachectic melanomas. The heatmaps are based on z-scores of group averages from baseline-adjusted log2(TMP) for genes with at least one significant call among the related comparison sets. **b** GO term analysis of genes that are significantly regulated (34 upregulated, 32 downregulated; $P < 0.05$ and 50% change; Supplementary Data 5) in the diaphragm of mice with cachectic melanomas, treated with rFibcd1 versus mock. **c** Expression of genes related to myofiber atrophy from dataset in **b** and Supplementary Data 5. Data are means ± SD. $P$ values were determined as reported in the RNA-seq methods and refer to cachectic versus non-cachectic, and to cachectic+rFibcd1 versus cachectic; *$P < 0.05$, **$P < 0.01$, ***$P < 0.001$. The N refers to diaphragm muscles and it is indicated in the legend of **a**. **d** Analysis of motor function in mice with either cachectic or non-cachectic orthotopic melanoma xenografts, and which received 3 i.p. injections of either rFibcd1 or mock (BSA) a week before testing. Melanoma-induced cachexia significantly reduces motor function compared to mice implanted with non-cachectic cancer cells. Comparison of the treadmill capacity of rFibcd1- versus mock-treated cachectic mice shows a trend towards improved treadmill performance in response to rFibcd1 injection but this difference does not reach statistical significance. However, cachectic mice treated with rFibcd1 have a treadmill capacity that is also not statistically different from that of non-cachectic mice. Data are means ± SD, with mice bearing non-cachectic melanomas and either mock-treated ($n = 7$) or treated with rFibcd1 ($n = 7$), and mice bearing cachectic melanomas and either mock-treated ($n = 8$) or treated with rFibcd1 ($n = 8$). $P$ values were determined by one-way ANOVA with Tukey's post hoc test; *$P < 0.05$ and **$P < 0.01$. Source data are provided in the Source Data file.

---

assayed 2 days after transfection. The following ON-TARGET plus siRNA reagents (Dharmacon) were used: mouse Fibcd1 (#L-0581480-01), non-targeting (NT) control (#D-001810-10), Itga2b (#L-046584-01), Wnt9a (#L-0581480-01), Tgfbi (#L-040939-03), Bmp1(#L-057019-01), and Sparc (#L-043364-00).

For concurrent treatment with rFibcd1, myotubes were then treated for 24 h with either serum-free media containing 100 ng/mL rFibcd1 or control serum-free media (containing the same amount of BSA as the medium with rFibcd1). Myotubes were then fixed and stained for myosin heavy chain (MF20 clone, eBioscience, #14-6503-82; used at 1:150) and myotube diameters analyzed by ImageJ, as explained in detail below. For the analysis of myoblast fusion shown in Supplementary Fig. 2, siRNAs were transfected into myoblasts following the procedures explained above.

**Nutrient-starvation experiments**. For nutrient starvation experiments, C2C12 myotubes were transfected as indicated above with Fibcd1 or NT siRNAs. After 48 h, the normal cell culture medium (10% FBS in DMEM-high glucose) was replaced with culture medium diluted 1:10 in Dulbecco's PBS for 8 h and 24h[168].

**Pharmacological inhibition of ERK**. C2C12 myotubes were transfected individually with Fibcd1 siRNA or NT siRNA for 48 h, as explained in detail above. The cells were incubated in serum-free media overnight and they were then treated at the same time with either control serum-free media or serum-free media containing rFibcd1 at 100 ng/mL, and/or Pyrazolylpyrrole (pharmacological inhibitor of ERK; Santa Cruz, CAS#933786-58-4)[35,85,86] at 2.5 ng/mL, for further 24 h. Cells were then fixed and stained with anti-Myosin Heavy Chain antibodies (MF20 clone, eBioscience, #14-6503-82; used at 1:150) and myotube size determined with ImageJ.

**C2C12 myotube size analysis**. To determine the myotube size, cultures of differentiated C2C12 myotubes were fixed by adding an equal volume of 4% PFA (paraformaldehyde, Electron Microscopy Sciences, #15710) to the medium for 10 min. Cells were then washed with PBS and blocked for 1 h in blocking buffer containing PBS with 0.1% Triton X-100 and 2% BSA, and then incubated with anti-myosin heavy chain antibodies (MF20 clone, eBioscience, #14-6503-82; used at 1:150) overnight at 4 °C. The cells were then stained with a fluorophore-conjugated anti-mouse secondary antibody to detect myosin heavy chain and with DAPI (Roche, #10236276001; 1:1000) to visualize the myotube area and the nuclei respectively. The size of myotubes was measured by taking the average width value across a myotube at three separate points along it. Typically, ~100 myotubes were measured for each group, as previously done[60].

**Induction of myotube atrophy via cachectic cytokines**. For the induction of myotube atrophy via cachectic cytokines, C2C12 myotubes were incubated in serum-free media for 6 h and then cachectic cytokines and rFibcd1 were added at the same time for a further 2 days. The cytokines used were IL-6 (recombinant mouse IL-6, R&D Systems, #406-ML-025) at 20 ng/mL[80], TNFα (recombinant mouse TNFα, R&D Systems, #410-ML-025) at 100 ng/mL[79], and LIF (recombinant mouse LIF, R&D Systems, #8878-lf-025) at 20 ng/mL[78]. Recombinant Fibcd1 (rFibcd1) was added to serum-free media at 10 ng/mL and 100 ng/mL, as indicated.

**Culture of cancer cells**. Cancer cells tested in Fig. 3 and Supplementary Fig. 3b are the following: the mouse B16F0 and B16F10 melanoma cells[169] (ATCC #CRL-6322 and #CRL-6475); the human MDA-MB-231 breast cancer cells[170] (ATCC #CRM-HTB-26); the mouse Ep5[171], Ep5ExTu[171] (Labelle lab collection), E0771[172] (CH3 BioSystems #94A001), 67NR[173] (F. Miller lab collection, Karmanos Cancer Institute), and 4T1[64,65] (ATCC #CRL-2539) breast cancer cells; the human LS180

colorectal adenocarcinoma cells[174] (ATCC #CL-187); the mouse MC38 colorectal adenocarcinoma cells[175] (D. Vignali lab collection, University of Pittsburgh); the human U-2OS-Luc/YFP[176], 143B-Luc/YFP[177], and Saos-2[92,178] osteosarcoma cells (ATCC #HTB-96, #CRL-8303, and #HTB-85); and the mouse Lewis lung carcinoma cells (LLC)[89] (ATCC, #CRL-1642).

The following culture media were used: RPMI-1640 (ATCC, #30-2001) for B16F0, B16F10, 67NR, and E0771; DMEM (ATCC, #30-2002) for MDA-MB-231, Ep5, Ep5ExTu, LS180, MC38, and LLC; EMEM (ATCC, #30-2003) for 143B-Luc/YFP; and McCoy's 5 A (ATCC, #30-2007) for U-2OS-Luc/YFP, 143B-Luc/YFP, and Saos-2. All culture media included 10% FCS (ATCC, #30-2020) with 1% penicillin/streptomycin (#15140122) apart for Saos-2 cells, which were cultured in 15% FCS, and for E0771 cells, which were cultured with 10% iron-supplemented FCS (GE Healthcare, HyClone, #SH30072.04HI). All cells were cultured at 37 °C with 5% $CO_2$.

All cell lines were screened regularly to ensure the absence of mycoplasma (universal mycoplasma detection kit, ATCC #30-1012 K) and authenticated based on their morphology and behavior. Human cell lines were also authenticated via STR (short tandem repeat) profiling.

**Treatment of cancer cells with rFibcd1**. Cancer cells were incubated in serum-free media overnight and they were then treated with rFibcd1 at 10 ng/mL and 100 ng/mL for 30 and 120 min when they were ~50% confluent. The cell lysate was then collected with NP40 buffer and used for western blotting.

**HEK293 cell culture and transfection**. Human embryonic kidney cells (HEK293T cells; ATCC, #CRL-3216) were cultured at 37 °C with 5% $CO_2$ in DMEM (high glucose DMEM, with Glutamax, GIBCO, #10566016) containing 10% fetal bovine serum (GIBCO, #10437-028), and penicillin/streptomycin (10,000 U/mL, GIBCO, #15140122). HEK293 cells were transfected with either empty vector (EV) or vector containing C-terminal FLAG tagged full length (FL) or short (SH) Fibcd1. 2 days after transfection, growth media was replaced with serum-free media for an additional 24 h. The media was then collected alongside the cell lysate using NP40 buffer and the samples run on SDS-PAGE and analyzed by western blot with anti-FLAG antibodies (Sigma clone M2, #A2220).

**Western blotting**. For cell culture experiments, the cells were homogenized in 100 μL of NP40 cell lysis buffer (Invitrogen, #FNN0021). Subsequently, the cell samples were sonicated for 10 sec at 30% of frequency, and the protein extracts were quantified by using Bradford assay (Bio-Rad Protein assay dye reagent concentrate, #5000006). Protein samples were prepared by addition of SDS-Blue loading buffer (Cell Signaling, #7722) and DTT (dithiothreitol; to a final concentration of 0.1 M, Cell Signaling, CST#1425 S) and heating at 95 °C for 5 min. Samples were then run on 4-20% gradient SDS-PAGE gels (Mini-PROTEAN TGX Pre-cast gels, Bio-Rad, 4561096) alongside a molecular weight ladder (Precision Plus protein standard, Bio-Rad, #1610374) and transferred to PVDF membranes (Immobilon-P PVDF membrane, Millipore, #IPVH00010), which were blocked with either 5% milk powder or 5% BSA (according to manufacturer's instructions) for 1 h.

Subsequently, the membranes were incubated overnight at 4 °C with the following primary antibodies (all used at 1:1000): mouse anti-phospho-p44/42 MAPK (Erk1/2; Thr202/Tyr204) (Cell Signaling, #9106), rabbit anti-phospho-p38 MAPK (Thr180/Tyr182) (Cell Signaling, #4511), rabbit anti-phospho-Smad2 (Ser465/467) (Cell Signaling, #8828), rabbit anti-phospho-Akt (Ser473; D9E) (Cell Signaling, #4060 S), rabbit anti-phospho-SAPK/JNK (Thr183/Tyr185; 81E11) (Cell Signaling, #4668 S), rabbit anti-α-Tubulin (11H10) (Cell Signaling, #2125), rabbit anti-SQSTM1/p62 (Cell Signaling, #5114), and mouse anti-Flag (M2) (Sigma,

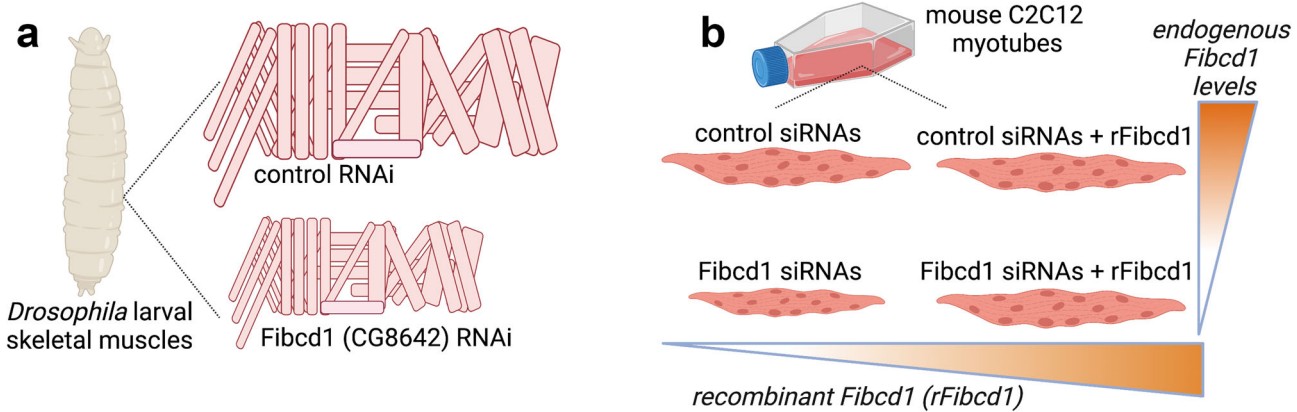

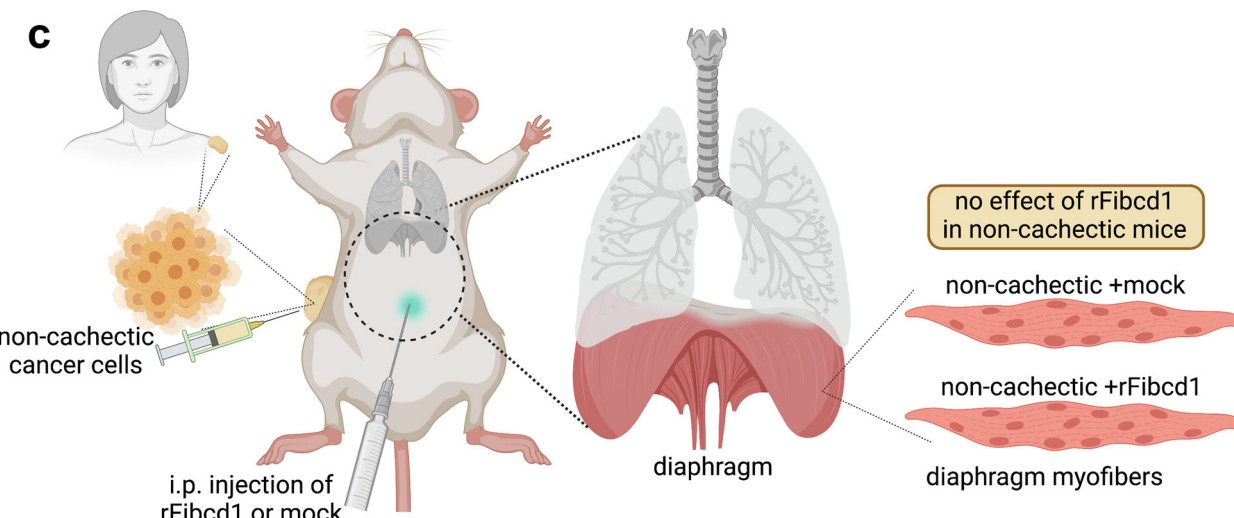

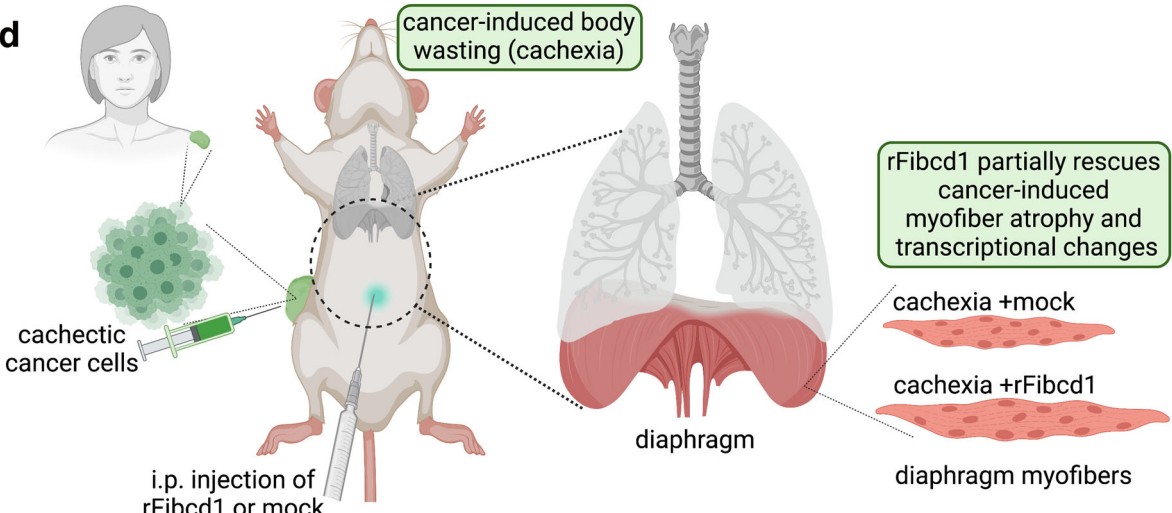

#F3165). After washing, the appropriate HRP-conjugated secondary antibodies (Cell Signaling #7074, #7076; used at 1:2000) were incubated at 4 °C for 2 h, washed again, and then probed with ECL reagents (Amersham ECL Western Blotting Detection Reagents, #RPN2209) to detect the protein of interest. Ponceau S (Sigma, #P7170) staining was also used to confirm even loading of protein samples. The Source Data file reports the full scan of all blots.

**qRT-PCR**. qRT-PCR was performed as previously described[35]. Total RNA from C2C12 myotubes was extracted by using Trizol (Ambion, #15596018). 500 ng of RNA was used for reverse transcription with the iScript cDNA synthesis kit (Bio-Rad, #1708840). qRT-PCR was done by using the IQ Sybr Green supermix (Bio-Rad, #170-8885) and the oligonucleotides reported in Supplementary Data 3. *Ppia* was used for normalization.

**Fig. 8 rFibcd1 partially rescues cancer-induced myofiber atrophy and transcriptional changes in the diaphragm muscle of cachectic mice. a** Fibcd1 (CG8642) was identified in a *Drosophila* RNAi screen as a myokine necessary for skeletal muscle growth. **b** Subsequent testing of mouse Fibcd1 in cultured C2C12 myotubes confirmed that Fibcd1 is necessary for muscle cell growth: siRNAs for Fibcd1 reduced myotube size and this was rescued by administration of recombinant Fibcd (rFibcd1). **c, d** Mice implanted with different pediatric melanoma xenografts. **c** A first xenograft consists of cancer cells that do not induce body wasting ("non-cachectic", i.e. cachexia-non-inducing): intraperitoneal (i.p.) injection of rFibcd1 does not impact myofiber size and transcriptional changes in the diaphragm muscle of these mice. **d** Profound (~25%) body weight loss (cachexia) is caused by a different melanoma xenograft ("cachectic", i.e. cachexia-inducing); cancer-induced myofiber atrophy and gene expression changes in the diaphragm are partially rescued by i.p. injection of rFibcd1. Similar results are also found in a cachexia model caused by subcutaneous injection of LLC cancer cells.

**Mouse husbandry**. All mice were housed in the Animal Resource Center at St. Jude Children's Research Hospital and handled in accordance with a protocol (#563) approved by the St. Jude Children's Research Hospital Institutional Animal Care and Use Committee (IACUC). Additional accreditation of the Animal Resource Center at St. Jude Children's Research Hospital was provided by the Association for Assessment and Accreditation of Laboratory Animal Care (AAA-LAC). Mice were housed in a ventilated rodent-housing system with a controlled temperature (22-23 °C), 40% humidity, 12-hour light/dark cycle, and given free access to food and water. Humane endpoints were not exceeded in any experiment.

**Mice that carry LLC (Lewis Lung Cancer) tumors**. Male C57BL/6 J (The Jackson Laboratory, JAX#000664) mice were utilized at 4 months of age. $1 \times 10^6$ LLC cells were each injected into the right and left flank[60,96,97] and tumors were allowed to grow up to ~3 weeks. Treatment with rFibcd1 began at day 16 after LLC tumor cell implantation. Specifically, mice were treated with three injections of recombinant Fibcd1 (rFibcd1) administrated intraperitoneally at 1 mg/kg of body weight or with 1%BSA in PBS on every other day for a week, at which time tumor-bearing mice were euthanized. Supplementary Fig. 9 reports a flow chart of the experimental design.

**Melanoma xenograft-bearing mice**. Female 2-month-old NCI Ath/nude mice were purchased from Charles River Laboratory. MAST360B/SJMEL030083_X2 and MAST552A/SJMEL031086_X1 cells were patient-derived melanoma cancer xenografts from the Childhood Solid Tumor Network collection at St. Jude Children's Research Hospital[111–115]. $1.5 \times 10^6$ cells of each tumor stock were inoculated into the flank of nude mice. Treatment with rFibcd1 began at day 39 after implantation of MAST360B/SJMEL030083_X2 ("cachectic", i.e. cachexia-inducing melanoma) and MAST552A/ SJMEL031086_X1 ("non-cachectic", i.e. cachexia-non-inducing melanoma). Mice were treated with three injections of recombinant Fibcd1 (rFibcd1) or with 1%BSA in PBS administrated intraperitoneally at 3 mg/kg of body weight on every other day for a week, after which tumor-bearing mice were euthanized. Supplementary Fig. 10 reports a flow chart of the experimental design.

**Tissue collection**. For tissue collection, mice were euthanized and the diaphragm muscle was dissected and isolated from tendons. Only the skeletal muscle portion of the dissected tissue was used for further analyses. Half of diaphragm muscle was frozen in liquid nitrogen-cooled isopentane (Sigma-Aldrich, #277258) and mounted for cryosectioning at a thickness of 10 μm whereas the other half was snap-frozen and stored at -80 °C until RNA extraction. The tumors were removed and weighed alongside.

**Analysis of myofiber type, size, and number**. Myofiber type analysis was done as previously described[179,180]. Unfixed slides holding the sections were incubated with blocking buffer (PBS with 2% BSA and 0.1% Triton-X100) for 1 h before incubation with primary antibodies (used at 1:150) overnight at 4 °C. The primary antibodies used were mouse IgG2b anti-myosin heavy chain type I (DSHB, #BA-F8), mouse IgG1 anti-myosin heavy chain type IIA (DSHB, #SC-71), and rat anti-laminin α2 (4H8-2; Santa Cruz, #sc-59854). The sections were then washed and incubated with secondary antibodies (1:200) which were anti-mouse IgM Alexa 555 (Life Technologies, A21426), anti-mouse IgG1 Alexa488 (Life Technologies, #A21121), and anti-rat IgG Alexa647 (Life Technologies, #A21247). Diaphragm sections were imaged on a Nikon C2 confocal microscope with a 10x objective and the myofiber types and sizes were analyzed in an automated manner with the Nikon Elements software by using the inverse threshold of laminin α2 immunostaining to determine myofiber boundaries. The myosin heavy chain staining was used to classify type I (red), type IIA (green), and presumed type IIX/IIB myofibers (black, i.e. that were not stained for MHC I or IIA). After myofibers were classified by type, myofiber size was estimated in an automated manner by the Nikon Elements software via the Feret's minimal diameter, a geometrical parameter for the analysis of unevenly shaped or cut objects[179]. For the quantification of the number of myofibers, all fibers in the diaphragm cross-sections were counted based on the myofiber borders identified by laminin α2 immunostaining.

**Analysis of motor function with the treadmill**. Mice were introduced onto the treadmill belt for acclimatization before the actual test (motor speed set to zero, for 5 min). Subsequently, the motor speed was set to 9 m/min in a flat position, and the speed was increased by 1 m/min every 2 min until the tested mouse was exhausted. Exhaustion was defined as the inability of the animal to run on the treadmill for 10 s. Instead of a shock stimulus, mice were prompted to run by using a tongue depressor, as previously done[136,137,181]. Running time and maximum speed were measured, whereas total running distance and average speed were calculated. Throughout this protocol, mice were assessed by an investigator blinded to the experimental conditions of the mice examined. No mice were excluded from the analysis.

**RNA-seq**. Diaphragm muscles were homogenized in Trizol and RNA extracted by isopropanol precipitation from the aqueous phase. RNA sequencing libraries for each sample were prepared with 1 μg total RNA by using the Illumina TruSeq RNA Sample Prep v2 Kit per the manufacturer's instructions, and sequencing was completed on the Illumina NovaSeq 6000. The 100-bp paired-end reads were trimmed, filtered against quality (Phred-like Q20 or greater) and length (50-bp or longer), and aligned to a mouse reference sequence GRCm38 (UCSC mm10) by using CLC Genomics Workbench v12.0.1 (Qiagen). For gene expression comparisons, we obtained the TPM (transcript per million) counts from the RNA-Seq Analysis tool. The differential gene expression analysis was performed by using the non-parametric ANOVA using the Kruskal-Wallis and Dunn's tests on log-transformed TPM values between biological replicates of experimental groups, implemented in Partek Genomics Suite v7.0 software (Partek Inc.).

The average TPM counts from each experimental group for significant genes (defined as indicated in the corresponding figure legends) were further clustered in a heatmap using z-score normalization and similarity measure by correlation. Specifically, for the heatmap in Fig. 4, the cutoff for significance is $p < 0.05$ and log2R > 0.585 (50% change) between these sets, and genes with significant call for at least one of the comparisons were included in the heatmap. The heatmaps in Fig. 5 includes all genes with significance in at least one of eight comparisons, with cutoffs of significance at p < 0.01 and log2R > 1. The heatmap in Supplementary Fig. 5 was generated following the same criteria as for Fig. 5. For the heatmap in Fig. 7, the cutoff of significance was set at p < 0.05 with log2R > 1 (one-fold change) for the comparison of CAC versus NCAC, and p < 0.05 with log2R > 0.585 (50% change) for the comparison of rFibcd1 versus mock treatment.

The RNA-seq data discussed in this publication has been deposited in the NCBI's Gene Expression Omnibus and is accessible through GEO Series accession numbers GSE156815, GSE158581, and GSE183833.

**Measurement of cytokine levels in the plasma of experimental mice**. The plasma cytokine levels were measured by using the Milliplex Map mouse cytokine assay kit (Millipore, #MCYTOMAG-70K) as previously done[182,183]. In total 600 μL of blood was collected from the abdominal aorta of mice that carry melanoma xenografts by using 100 μL of 50 mM EDTA as an anti-coagulant and centrifuged for 20 min at 1,000 g. All reagents were brought to room temperature before use. Wash buffer, assay buffer, serum matrix, standard 6, quality controls 1 and 2, premixed beads, detection antibodies, and streptavidin-PE were prepared as recommended by the manufacturer. Plasma samples were diluted 1:4 by using the assay buffer. To create a homogeneous mixture, the premixed beads bottle was sonicated in water bath for 30 s and then vortexed for 1 min before use. For pre-wetting, 200 μL assay buffer was pipetted into each well, the plate was covered with sealer and then shaken at 70 g for 10 min. The fluid was removed by tapping the plate on a paper towel and by centrifuging it briefly at 1,200 g lying top down on a paper sheet in the centrifuge. A total of 25 μL of standards 1–6 and quality controls 1 and 2 were pipetted in duplicate into the appropriate wells and 25 μL of serum matrix was added. Next, 25 μL of diluted plasma samples and 25 μL of assay buffer was pipetted into the appropriate wells. Finally, the magnetic premixed beads were vortexed for 1 min and 25 μL was pipetted into each well. The plate was sealed, covered with aluminum foil and then shaken at 70 g for 16 h at 4 °C. After overnight incubation, the magnetic bead plates were washed 2 times by using assay buffer, as recommended by the manufacturer. Then, 25 μL of detection antibodies were added into each well, the plate was sealed and covered with aluminum foil and shaken at 70 g for 1 h. After that, 25 μL of streptavidin-phycoerythrin was added to each well containing the 25 μL of detection antibodies. The plate was again sealed, covered with foil and incubated with agitation for 30 min at room temperature. Subsequently, the plate was washed 2 times again. In 150 μL of assay buffer were added to all wells and shaken for 5 min to resuspend the beads. Lastly, the plate was read on a Luminex® 200™ apparatus. The analysis was done by the Luminex

software by using the spline curve-fitting method for calculating cytokine/che-mokines concentrations in samples.

**Reporting summary**. Further information on research design is available in the Nature Research Reporting Summary linked to this article.

## Data availability

The RNA-seq data discussed in this publication have been deposited in the NCBI's Gene Expression Omnibus and are publicly accessible through GEO Series accession numbers GSE156815, GSE158581, and GSE183833.

Additional primary data are available in Supplementary Data 1–5 and in the Source Data File.

Publicly-available datasets that have been interrogated in this study are the following and were used to retrieve RNA-seq data from pediatric cancers (ProteinPaint; https://proteinpaint.stjude.org/) and adult cancers (FireBrowse; http://firebrowse.org/viewGene.html?gene=ITGA2B), and for assessing circulating levels of Fibcd1 (Aging Plasma Proteome dataset; https://twc-stanford.shinyapps.io/aging_plasma_proteome/).

**Statistics and Reproducibility**. Data organization, scientific graphing and statistical analyses were performed with Microsoft Excel (version 14.7.3) and GraphPad Prism (version 6). The unpaired two-tailed Student's $t$-test was used to compare the means of two independent groups to each other. One-way and two-way ANOVA with post hoc testing were used for multiple comparisons of more than two groups of normally distributed data. The "n" for each experiment can be found in the figure legends and represents independently generated samples for all experiments, including myotubes and cell culture samples for in vitro assays, and individual animals or tissues sourced from independent animals for in vivo experiments. Bar graphs present the mean ± SD. Throughout the figures, asterisks and ampersand symbols indicate the significance of $P$ values: *$P < 0.05$, **$P < 0.01$, ***$P < 0.001$, ****$P < 0.0001$. A significant result was defined as $P < 0.05$. Representative micrographs are derived from the analysis of multiple muscles (>5) and cell culture results typically obtained from 2-3 experiments.

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

## Acknowledgements

Fly stocks were provided by the VDRC, NIG-Fly, and Bloomington stock centers whereas the DSHB provided MHC antibodies. We thank Dr. Richard Heath and the Protein Production core, the Light Microscopy facility, and the Hartwell Center for Bioinformatics and Biotechnology at St. Jude Children's Research Hospital. We further thank Åsa Karlström and the CIVIT at St. Jude Children's Research Hospital for assistance with xenograft experiments. This work was supported primarily by a research grant to F.D. from The Hartwell Foundation (Individual Biomedical Research award). F.D. is supported also by the National Cancer Institute (P30CA021765-developmental funds) and the National Institute on Aging (R01AG055532 and R56AG63806). B.X. and Y.F. are supported by P30CA021765. M.L. is supported by the National Cancer Institute (R01CA245301). L.C.H. was supported by a Glenn/AFAR Postdoctoral Fellowship. Research at St. Jude Children's Research Hospital is supported by the ALSAC. The content is solely the responsibility of the authors and does not necessarily represent the official views of the National Institutes of Health.

## Author contributions

F.A.G. did most of the cell culture and mouse experiments, with help from L.C.H. and A.S.; M.R. did the *Drosophila* RNAi screen; F.A.G., M.R., and L.C.H. analyzed data; Y.-D. W. analyzed RNA-seq data; B.X. and Y.F. provided support for other computational analyses; F.A.G. and G.Q. measured circulating cytokines; B.G. established and provided cancer xenografts; M.L. and R.W. provided expertise and tools for experiments with cancer cell lines; F.A.G. is the first author; M.R. and L.C.H. equally contributed as second authors; F.D. supervised the project and wrote the manuscript.

## Competing interests

F.A.G., M.R., L.C.H., and F.D. are named co-inventors of a pending U.S. provisional patent application based in part on the research reported in this paper. The remaining authors declare no competing interests.
