## [Peer Review File · Nature Communications]

Reviewers' Comments:

Reviewer #1:

Remarks to the Author:

The current investigation examines FIBCD1's role in muscle, specifically the diaphragm. The study provides data on a novel and innovative screen of myokines in drosophila muscle growth to identify Fibcd1. The study defines the role of Fibcd1 to regulate muscle mass through MAPK / ERK signaling.

Novel therapeutic targets for the prevention or treatment of cachexia are needed and provide high significance. Defining the role of FIBCD1 as a myokine and in skeletal muscle regulation is novel and important. The study's strength lies in the identification of FIBCD1 in a drosophila muscle growth screen and the rigorous characterization of its role in vitro related to C2C12 myotube growth regulation and cytokine-induced atrophy. Furthermore, the study contains an examination of rFobsc1 rescue in vivo using the established LLC cachexia model and a less characterized melanoma xenograft model.

FIBCD1 expression in tissues has been investigated with cancer, and high expression level in tissues has been associated with poor prognosis and decreased survival. Discussion of this dichotomy is warranted.

While the diaphragm was examined, cancer-induced wasting is thought to at least initially be more prominent in glycolytic myofibers. Do the investigated effects extend beyond the oxidative and high-use diaphragm muscle?

While myokines are discussed, the experimental approaches deal with systemic administration. Please highlight sources of FIBCD1 expression in over tissues. This emphasis should be tempered in the study unless the myokine role can be more extensively examined.

For the rFibcd1 in vivo experiment, please clearly define "treating" the cachectic condition versus prevention. Both are important but distinct. Current research has moved to reverse the cachectic phenotype, which is not demonstrated with the current data set. Please correct.

Tumor size is not the only indicator of the cachectic potential of a tumor. Very small tumors cause cachexia. Please edit that effects on tumor size are the only indicator of effects that influence the systemic cancer environment that is conducive to wasting. Furthermore, there are many references in the text to cancer cells when the "tumor" microenvironment consists of multiple cell types that can impact the systemic environment. This needs to be updated in the text.

The variable nature of cachexia with the LLC model requires some indication of the degree of wasting in the mice that the diaphragms were taken.

The melanoma model is not well characterized as a cachexia model. Additional information on the cachectic state of the mouse related to standard variables of muscle and fat mass is warranted.

Please discuss that muscle ERK is activated by hypertrophy and activated by many cachectic stimuli, and what are the implications of this dual activation.

Introduction 2nd paragraph states the role of cancer cell secretes cytokines. Please reflect on current work on the role of immune cell secretion of cytokines and other factors for changes in the systemic environment conducive to wasting.

Introduction 3rd paragraph, the role of myokines, and the myostatin example are superficial. Please bolster the presentation to represents the array of myokines that can myokine regulate muscle mass.

The discussion needs to be reformatted to condense several 2 and 3 sentence mini paragraphs into a more coherent scientific discourse related to the main findings.

Reviewer #2:

Remarks to the Author:

This is an extremely well-written paper that reports the identification of a myofiber size regulatory myokine in *Drosophila* that is later characterized in C2C12 cells and in the mouse in a cancer-induced diaphragm caquexia model. Despite the authors do a great job characterizing the activity of Fibcd1 in vivo, its mechanism of rescue, histological rescue and transcriptomics, among other results, it remains to be demonstrated whether any of those rescues translate into a real improvement in diaphragm (respiratory) function. Assessment of breathing capacity would make the paper outstanding but perhaps other, less technically demanding approaches, would suffice such as extending the model mouse half-life, increasing exercise endurance or the like.

As for other questions that require attention by the authors I summarize the following:

1. The statement that "rFibcd1 does not promote ERK signaling in a set of cancer cells" is probably too bold given the relatively reduced number of cancer cell lines that are actually tested. Also, the number of technical repeats is 2, and the activation of ERK signaling in Saos cells 30 min post-administration is buried in the text but looks significant. Thus, being the idea of a muscle-specific treatment central to the paper, I believe these experiments require additional support.

2.- The conclusion that integrin alpha2b is required for the differential response is appealing but it is only formally demonstrated for the muscle response to the myokine, not for the lack of sensitivity of the cancer cells. For example, Saos cells have low integrin expression but do seem to respond to the myokine. Thus, either the text is modified or more evidence is provided, for example, introduce the integrin and gain sensitivity to the treatment.

3.- Please discuss potential explanations why the myokine also activates ERK in control conditions (NT siRNA) despite rFobcd1 does not enhance myotube size under such conditions.

4.- Fig. 3b, diameter is 40 microns while it was previously 20. Please comment.

5.- Fig. 4ij. Figure legend states n=8 but graphs include only 5 data points. Same with orange points. Please explain.

6.- Fig. 5 and 7, it is unclear why only a specific number of genes are included in the heatmaps. How are 2043 or 841 genes selected? What criteria do they meet?

Minor points:

1.- It is unclear how many technical repeats in the siRNA experiment with the alpha 2b integrin in C2C22 cells.

2.- Please include a space between values and units.

3. Fig. 3b, detection of tubulin in a protein extract from cell culture supernatants?

4.- Supp Fig 2. Some differences shown in panel (a) are for sure significant but are not labeled accordingly.

Reviewer #3:

Remarks to the Author:

Very nice study. I am favorably impressed.

Just a few comments:

1. A flow diagram (like in human studies, especially RCTs) would be nice to show how and why the authors ended up with each given number of mice for each outcome/experiment. The reader needs to be convinced that the authors used no 'convenience' sample/s. This statistical/methodological tool is seldom used in preclinical/mechanistic studies as the present one and this is unfortunate I think.

2. I am missing a cartoon (infographic abstract) summarizing the study design and its main results

3. Abstract: 'However, no therapies are available'. What about specific inspiratory muscle training? We and many others have successfully used this intervention in chronic disorders in which ventilatory muscles are even more affected than in cancer cachexia (e.g., mitochondrial disorders). I doubt there is any skeletal muscle that is non-responsive to specific exercise training, especially if we are speaking of the benefits of myokines! I have done too many studies in mice and humans to think that a muscle cannot improve its size and function with the proper exercise stimulus.

4. Maybe the concept of 'myokines' is a bit restrictive now and we should rather speak about 'exerkines' (i.e., molecules (chemokines, cytokines and small peptides in general but also other types of molecules) released in (and specific to) the exercise milieu but not necessarily derived solely from contracting muscle fibers.

RESPONSE TO THE REVIEWERS' COMMENTS

REVIEWER COMMENTS

Reviewer #1 (Remarks to the Author):

The current investigation examines FIBCD1's role in muscle, specifically the diaphragm. The study provides data on a novel and innovative screen of myokines in drosophila muscle growth to identify Fibcd1. The study defines the role of Fibcd1 to regulate muscle mass through MAPK / ERK signaling.

Novel therapeutic targets for the prevention or treatment of cachexia are needed and provide high significance. Defining the role of FIBCD1 as a myokine and in skeletal muscle regulation is novel and important. The study's strength lies in the identification of FIBCD1 in a drosophila muscle growth screen and the rigorous characterization of its role in vitro related to C2C12 myotube growth regulation and cytokine-induced atrophy. Furthermore, the study contains an examination of rFibcd1 rescue in vivo using the established LLC cachexia model and a less characterized melanoma xenograft model.

Thank you for the positive evaluation of our work.

FIBCD1 expression in tissues has been investigated with cancer, and high expression level in tissues has been associated with poor prognosis and decreased survival. Discussion of this dichotomy is warranted.

Response: Thank you for pointing this out. We have found that rFibcd1 reduces the expression of inflammatory genes (which are a major driver of wasting) and myofiber atrophy in the diaphragm skeletal muscle. Consistent with a role of Fibcd1 in reducing inflammation, a recent study has found that gut-specific Fibcd1 overexpression reduces intestinal inflammation and impedes weight loss induced by chemotherapy in mice. Taken together, our study and this recent publication suggest that Fibcd1 reduces inflammation. However, paradoxically, high expression of FIBCD1 in cancer cells has been found to correlate with poor prognosis and survival in patients with gastric and liver cancer. A possible explanation for this dichotomy is that FIBCD1 expression by cancer cells impedes local inflammatory responses and hence allows cancer immune escape, which is a key mechanism of cancer progression and metastatic dissemination. We have now better discussed this dichotomy in a new paragraph in the Discussion.

While the diaphragm was examined, cancer-induced wasting is thought to at least initially be more prominent in glycolytic myofibers. Do the investigated effects extend beyond the oxidative and high-use diaphragm muscle?

Response: Our investigation and i.p. injection of rFibcd1 specifically target the diaphragm muscle because of previous studies highlighting the importance of myofiber atrophy in the diaphragm for determining cachexia-associated mortality in cancer patients. In the diaphragm, we find that rFibcd1 partially rescues myofiber atrophy induced by cachexia in all myofiber types (type 1, 2a, and 2x/2b), both oxidative and glycolytic.

While myokines are discussed, the experimental approaches deal with systemic administration. Please highlight sources of FIBCD1 expression in other tissues. This emphasis should be tempered in the study unless the myokine role can be more extensively examined.

Response: As the reviewer points out, our study examines only local effects on the diaphragm muscle deriving from i.p. injection of rFibcd1. Consequently, we have toned down the discussion of possible systemic effects that may derive from the systemic action of rFibcd1.

For the rFibcd1 in vivo experiment, please clearly define "treating" the cachectic condition versus prevention. Both are important but distinct. Current research has moved to reverse the cachectic phenotype, which is not demonstrated with the current data set. Please correct.

Response: Thank you for pointing this out. rFibcd1 injection is done at a late time point (i.e. a week before euthanasia). On this basis, we agree with the reviewer that rather than preventing cachexia our data are consistent with an effect of rFibcd1 in treating/rescuing cachexia-induced changes. We have consequently amended the manuscript.

Tumor size is not the only indicator of the cachectic potential of a tumor. Very small tumors cause cachexia. Please edit that effects on tumor size are the only indicator of effects that influence the systemic cancer environment that is conducive to wasting. Furthermore, there are many references in the text to cancer cells when the “tumor” microenvironment consists of multiple cell types that can impact the systemic environment. This needs to be updated in the text.

Response: We have amended the manuscript to better explain these important points. In particular, a new paragraph in the introduction now discusses that tumor burden and cachectic potential can be disconnected, and that the cachectic potential also relies on host factors and on the panel of cachectic cytokines produced by cancer cells and associated stromal cells.

The variable nature of cachexia with the LLC model requires some indication of the degree of wasting in the mice that the diaphragms were taken.

Response: Fig. 4i reports information of the degree of body wasting in the mice carrying ulcerated LLC tumors from which the diaphragms were taken: LLC cancers induce a ~10% decline in tumor-free body weight in this cachexia model.

The melanoma model is not well characterized as a cachexia model. Additional information on the cachectic state of the mouse related to standard variables of muscle and fat mass is warranted.

Response: A new figure (Figure 5) now reports information on the progressive development of diaphragm cachexia in this new melanoma model. Specifically, we have analyzed the progressive development of myofiber atrophy and transcriptional changes that occur in the diaphragm at different time points of cachexia progression. Moreover, Supplementary Figure 5 now reports how melanoma-induced cachexia impacts the mass of different tissues/organs (brown and white adipose, liver, pancreas, ovaries, heart, and skeletal muscles), the transcriptional changes that occur in different skeletal muscles (diaphragm, tibialis anterior, soleus), and the impact of cachexia on body weight.

Please discuss that muscle ERK is activated by hypertrophy and activated by many cachectic stimuli, and what are the implications of this dual activation.

Response: We have now better explained in the Discussion the possible implications of Erk activation by both hypertrophic stimuli and cachexia.

Introduction 2nd paragraph states the role of cancer cell secretes cytokines. Please reflect on current work on the role of immune cell secretion of cytokines and other factors for changes in the systemic environment conducive to wasting.

Response: We have now included this information in the Introduction.

Introduction 3rd paragraph, the role of myokines, and the myostatin example are superficial. Please bolster the presentation to represent the array of myokines that can myokine regulate muscle mass.

Response: A new paragraph in the introduction now better explains the role of myokines in regulating myofiber size in response to exercise and other physiological stimuli, as well as the possibility that myokines could be modulated to counteract disease-associated wasting.

The discussion needs to be reformatted to condense several 2 and 3 sentence mini paragraphs into a more coherent scientific discourse related to the main findings.

Response: Thank you for pointing this out. We have revised the discussion accordingly.

Reviewer #2 (Remarks to the Author):

This is an extremely well-written paper that reports the identification of a myofiber size regulatory myokine in *Drosophila* that is later characterized in C2C12 cells and in the mouse in a cancer-induced diaphragm cachexia model. Despite the authors do a great job characterizing the activity of Fibcd1 in vivo, its mechanism of rescue, histological rescue and transcriptomics, among other results, it remains to be demonstrated whether any of those rescues translate into a real improvement in diaphragm (respiratory) function. Assessment of breathing capacity would make the paper outstanding but perhaps other, less technically demanding approaches, would suffice such as extending the model mouse half-life, increasing exercise endurance or the like.

Response: We thank the reviewer for the positive evaluation of our work. We have now tested motor function with the treadmill in mice with either cachectic or non-cachectic orthotopic melanoma xenografts, and which received 3 i.p. injections of either rFibcd1 or mock (BSA) a week before testing. Melanoma-induced cachexia led to significant decline in motor function compared to mice implanted with non-cachectic cancer cells, and this was attenuated by rFibcd1 injection i.p., presumably because of improved diaphragm function. Overall, these findings suggest that rFibcd1-mediated improvements in cancer-induced myofiber atrophy and transcriptional changes of the diaphragm can translate into functional benefits. This new data is now shown in Fig. 7d. A new method paragraph describes how the treadmill assay was done.

As for other questions that require attention by the authors I summarize the following:

1. The statement that "rFibcd1 does not promote ERK signaling in a set of cancer cells" is probably too bold given the relatively reduced number of cancer cell lines that are actually tested. Also, the number of technical repeats is 2, and the activation of ERK signaling in Saos cells 30 min post-administration is buried in the text but looks significant. Thus, being the idea of a muscle-specific treatment central to the paper, I believe these experiments require additional support.

Response: Thank you for pointing this out, we have now revised the text and provided additional data. Specifically, Supplementary Fig. 3b reports the outcome of rFibcd1 treatment on a panel of additional cancer cell lines. We find that most cancer cell lines do not display increased P-ERK levels upon rFibcd1 treatment. There are however a few exceptions, including the 67NR cell line, which strongly responds to rFibcd1 (Fig. 3d and Supplementary Fig. 3b). Interestingly, compared to rFibcd1-non-responsive cancer cell lines, 67NR cells display higher Itga2b expression, at levels similar to those of C2C12 myotubes (Fig. 3f). These findings suggest that Itga2b might indeed be a receptor for rFibcd1. However, we acknowledge that additional receptors may also be involved and explain the response of Saos-2 cells to high concentrations of rFibcd1.

2.- The conclusion that integrin alpha2b is required for the differential response is appealing but it is only formally demonstrated for the muscle response to the myokine, not for the lack of sensitivity of the cancer cells. For example, Saos cells have low integrin expression but do seem to respond to the myokine. Thus, either the text is modified or more evidence is provided, for example, introduce the integrin and gain sensitivity to the treatment.

Response: We have changed the text to acknowledge that although integrin alpha2b is needed for the response to Fibcd1 in muscle cells, there could also be alternative rFibcd1 receptors.

3.- Please discuss potential explanations why the myokine also activates ERK in control conditions (NT siRNA) despite rFibcd1 does not enhance myotube size under such conditions.

Response: We speculate that in control conditions there could be other factors independent from Erk signaling that are limiting for additional growth. Our findings are indeed in line with the previous literature that has shown that Erk signaling is primarily needed for the maintenance of myofiber size rather than for myofiber hypertrophy (refs. 61-63). This point is now better explained in the Discussion.

4.- Fig. 3b, diameter is 40 microns while it was previously 20. Please comment.

Response: This mistake (due to an incorrect definition of the magnification in the image analysis software) has been corrected. We confirm that ~20 microns is the correct average size as reported in the other figures.

5.- Fig. 4ij. Figure legend states n=8 but graphs include only 5 data points. Same with orange points. Please explain.

Response: Thank you for pointing this out, this typo in the figure legend has now been corrected.

6.- Fig. 5 and 7, it is unclear why only a specific number of genes are included in the heatmaps. How are 2043 or 599 genes selected? What criteria do they meet?

Response: Thank you for pointing this out, we have now added the following additional information to the methods section of the manuscript:

The average TPM counts from each experimental group for significant genes (defined as indicated in the corresponding figure legends) were further clustered in a heatmap using z-score normalization and similarity measure by correlation. Specifically, for the heatmap in Fig. 4, the cutoff for significance is $p < 0.05$ and $\log_2 R > 0.585$ (50% change) between these sets, and genes with significant call for at least one of the comparisons were included in the heatmap. The heatmaps in Fig. 5 includes all genes with significance in at least one of eight comparisons, with cutoffs of significance at $p < 0.01$ and $\log_2 R > 1$. The heatmap in Supplementary Fig. 5 was generated following the same criteria as for Fig. 5. For the heatmap in Fig. 7, the cutoff of significance was set at $p < 0.05$ with $\log_2 R > 1$ (one-fold change) for the comparison of CAC versus NCAC, and $p < 0.05$ with $\log_2 R > 0.585$ (50% change) for the comparison of rFibcd1 versus mock treatment.

Minor points:

1.- It is unclear how many technical repeats in the siRNA experiment with the alpha 2b integrin in C2C12 cells.

Response: Consistent results have been obtained with 2 doses of rFibcd1 (10 and 100 ng/mL) and 30-min and 120-min treatments (Fig. 3g).

2.- Please include a space between values and units. **Done.**

3. Fig. 3b, detection of tubulin in a protein extract from cell culture supernatants?

Response: Fig. 3b reports the analysis of cell lysates and supernatants of cell transfected with FLAG-tagged Fibcd1 variants. The coomassie blue staining is shown as loading control whereas tubulin is shown to indicate the purity of the preparation. As expected, tubulin is abundantly detected in the cell lysate whereas minimal tubulin is detected in the conditioned media (this residual tubulin may arise from few cells that detached), thus serving as a quality control for the preparation of this fraction.

4.- Supp Fig 2. Some differences shown in panel (a) are for sure significant but are not labeled accordingly.

Response: In Supplementary Figure 2, there is no statistical significance unless already indicated due to the high standard deviation, despite substantial changes in the mean values of some groups. Nonetheless, the precise P-values are now reported in Supplementary Figure 2 for the graphs with major differences shown in (a).

Reviewer #3 (Remarks to the Author):

Very nice study. I am favorably impressed.
Thank you for the positive evaluation of our work.

Just a few comments:

1. A flow diagram (like in human studies, especially RCTs) would be nice to show how and why the authors ended up with each given number of mice for each outcome/experiment. The reader needs to be convinced that the authors used no 'convenience' sample/s. This statistical/methodological tool is seldom used in preclinical/mechanistic studies as the present one and this is unfortunate I think.

Response: Thank you for this suggestion. We have now included flow diagrams to better explain the study design (Supplementary Fig.8-9). Mice were randomly allocated into treatment groups (mock versus rFibcd1). In experiments with mice implanted with LLC cancer cells, a total of 10 mice were excluded because they carried non-ulcerated tumors, which develop minimal cachexia. In experiments with melanoma xenografts, 7 mice with cachectic melanoma xenografts were excluded from treatment because they developed minimal cachexia, as indicated by a decline in body weight of less than 10% and/or limited tumor burden (<1.3 cm tumor diameter). An investigator blinded to the experimental conditions assessed the result of rFibcd1 versus mock treatment. These diagrams are now shown in Supplementary Fig.8-9.

2. I am missing a cartoon (infographic abstract) summarizing the study design and its main results.

Response: Figure 8 now reports a cartoon that summarizes key findings of the study.

3. Abstract: 'However, no therapies are available'. What about specific inspiratory muscle training? We and many others have successfully used this intervention in chronic disorders in which ventilatory muscles are even more affected than in cancer cachexia (e.g., mitochondrial disorders). I doubt there is any skeletal muscle that is non-responsive to specific exercise training, especially if we are speaking of the benefits of myokines! I have done too many studies in mice and humans to think that a muscle cannot improve its size and function with the proper exercise stimulus.

Response: We apologize for this oversight. We have now amended the introduction to discuss the fact that although there are no pharmaceutical treatments, inspiratory muscle training has been found to improve diaphragm function in a number of disease conditions. We have also amended the discussion to indicate that rFibcd1 and pharmaceutical interventions could be used in conjunction with inspiratory muscle training to contrast diaphragm atrophy.

4. Maybe the concept of 'myokines' is a bit restrictive now and we should rather speak about 'exerkines' (i.e., molecules (chemokines, cytokines and small peptides in general but also other types of molecules) released in (and specific to) the exercise milieu but not necessarily derived solely from contracting muscle fibers.

Response: We agree that also secreted factors released in response to exercise (i.e. exerkines) contribute to regulate myofiber size and have amended accordingly the paragraph in the Introduction that discusses about secreted factors that regulate myofiber size.

Reviewers' Comments:

Reviewer #1:

Remarks to the Author:

The authors have been very responsive to prior critiques in editing the resubmitted manuscript. The edits have further enhanced the impact on the cancer cachexia field. However, related to some of the revisions and current presentation, a couple of minor issues were identified involving the interpretation of the treadmill test for systemic functional change and the evidence for the specificity of the myokine response to the diaphragm muscle

Comments

Figure 7d and associated text. Involving the additional functional treadmill tests, please clarify in the text if there was a difference between the cachectic melanoma run performance and the performance of the cachectic melanoma mice receiving rFibcd1. If there was no difference between these groups, please alter the conclusions on treadmill function or power the study to examine these differences more rigorously.

Results. >> The capacity of rFibcd1 to limit myofiber atrophy in the diaphragm of cancer-bearing mice arises via its local action in contrasting cachectic changes in the diaphragm muscle. Systemic effects of rFibcd1 on body wasting and/or on cancer cells are unlikely given that rFibcd1 is injected locally in the peritoneal cavity. Indeed, body weight and tumor-free body mass declined in mice injected with LLC cancer cells but did not differ in response to intraperitoneal rFibcd1 injection (Fig. 4h-i)<< This point is interesting but would be better established with examples of hindlimb muscle weights that did or did not change. If this is a critical aspect of the study, please supply additional data to support the muscle-specific action. This could be added to the text.

Prior comments on reformatting and focusing the discussion on removing 2& 3 sentence paragraphs were largely ignored. It is quite challenging to discuss a critical point with the needed depth in 2-3 sentences, which points to a more focused and in-depth discussion focusing on the central issues.

Minor

The new sentence >> The capacity of rFibcd1 to limit myofiber atrophy in the diaphragm of cancer-bearing mice arises via its local action in contrasting cachectic changes in the diaphragm muscle<< Clarity; possible change contrasting to attenuating (mitigating) or something similar

Reviewer #2:

Remarks to the Author:

The authors have satisfactorily addressed all my concerns. I found it particularly reassuring that all the molecular and histological rescues the authors report translate into mitigation of a functional phenotype as relevant as physical activity endurance (treadmill activity). As a very minor point, please note that Supp Fig. 3a shows the labels truncated and bar graphs of panel b are too small for comfortable reading.

Reviewer #3:

Remarks to the Author:

Thanks for your changes

RESPONSE TO THE REVIEWERS' COMMENTS

Reviewer #1 (Remarks to the Author):

The authors have been very responsive to prior critiques in editing the resubmitted manuscript. The edits have further enhanced the impact on the cancer cachexia field. However, related to some of the revisions and current presentation, a couple of minor issues were identified involving the interpretation of the treadmill test for systemic functional change and the evidence for the specificity of the myokine response to the diaphragm muscle.

Response: Thank you for the positive evaluation of our work. We have revised the manuscript according to the interpretation issues that were identified.

Comments

Figure 7d and associated text. Involving the additional functional treadmill tests, please clarify in the text if there was a difference between the cachectic melanoma run performance and the performance of the cachectic melanoma mice receiving rFibcd1. If there was no difference between these groups, please alter the conclusions on treadmill function or power the study to examine these differences more rigorously.

Response: Thank you for pointing this out. Data in Figure 7d indicates that cachexia significantly reduces treadmill performance, compared to non-cachectic controls. By comparing the treadmill capacity of rFibcd1 versus mock-treated cachectic mice, we find that there is a trend towards improved treadmill performance in response to rFibcd1 injection but this difference does not reach statistical significance. However, cachectic mice treated with rFibcd1 have a treadmill capacity that is also not statistically different from that of non-cachectic mice. We have now amended the results section of the manuscript and the figure legend to report more precisely the outcome of these studies. Figure 7d displays the precise P-values for all statistical comparisons.

Results. >> The capacity of rFibcd1 to limit myofiber atrophy in the diaphragm of cancer-bearing mice arises via its local action in contrasting cachectic changes in the diaphragm muscle. Systemic effects of rFibcd1 on body wasting and/or on cancer cells are unlikely given that rFibcd1 is injected locally in the peritoneal cavity. Indeed, body weight and tumor-free body mass declined in mice injected with LLC cancer cells but did not differ in response to intraperitoneal rFibcd1 injection (Fig. 4h-i)<< This point is interesting but would be better established with examples of hindlimb muscle weights that did or did not change. If this is a critical aspect of the study, please supply additional data to support the muscle-specific action. This could be added to the text.

Response: In support to this point, we now report data on the mass of the tibialis anterior muscle in cachectic and non-cachectic mice, treated with intraperitoneal injection of rFibcd1 or mock. Similar to the lack of changes in tumor-free body mass observed in response to rFibcd1, the weight of the tibialis anterior muscle does not change upon intraperitoneal injection of rFibcd1. This data is shown in Supplementary Figure 6.

Prior comments on reformatting and focusing the discussion on removing 2& 3 sentence paragraphs were largely ignored. It is quite challenging to discuss a critical point with the needed depth in 2-3 sentences, which points to a more focused and in-depth discussion focusing on the central issues.

Response: We have now tried to streamline the discussion by removing minor points that were not covered in depth because of their relative marginality to the central issues.

Minor

The new sentence >> The capacity of rFibcd1 to limit myofiber atrophy in the diaphragm of cancer-bearing mice arises via its local action in contrasting cachectic changes in the diaphragm muscle<< Clarity; possible change contrasting to attenuating (mitigating) or something similar

Response: Thank you for pointing this out, this has been changed.

Reviewer #2 (Remarks to the Author):

The authors have satisfactorily addressed all my concerns. I found it particularly reassuring that all the molecular and histological rescues the authors report translate into mitigation of a functional phenotype as relevant as physical activity endurance (treadmill activity). As a very minor point, please note that Supp Fig. 3a shows the labels truncated and bar graphs of panel b are too small for comfortable reading.

Response: Thank you for pointing this out. We have fixed the truncated label in Supp Fig. 3a. We now display Supp Fig. 3b over two pages for a more comfortable reading.

Reviewer #3 (Remarks to the Author):

Thanks for your changes.

Response: Thank you for your suggestions and for the positive evaluation of our work.